

# Relationship between microstructures and resistance in mafic assemblages that deform and transform.

Nicolas Mansard[1,*], Holger Stünitz[1,2], Hugues Raimbourg[1], Jacques Précigout[1], Alexis Plunder[3] and Lucille Nègre[1]

[1] Institut des Sciences de la Terre d'Orléans (ISTO), UMR 7327, CNRS/BRGM, Université d'Orléans, 45071 Orléans, France
[2] Department of Geology, University of Tromsø, Dramsveien 201, 9037 Tromsø, Norway
[3] BRGM, F-45060, Orléans, France

*Correspondence to*: Nicolas Mansard (nmansard@outlook.fr)

E-mail addresses: nmansard@outlook.fr (N. Mansard), holger.stunitz@uit.no (H. Stünitz), hugues.raimbourg@univ-orleans.fr (H. Raimbourg), jacques.precigout@univ-orleans.fr (J. Précigout), a.plunder@brgm.fr (A. Plunder), lucille.negre@univ-orleans.fr (L. Nègre).

**Abstract.**

Syn-kinematic mineral reactions play an important role for the mechanical properties of polymineralic rocks. Mineral reactions
(i.e. nucleation of new phases) may lead to grain size reduction producing fine-grained polymineralic mixtures, which have a strongly reduced viscosity because of the activation of grain-size sensitive deformation processes. In order to study the effect of deformation-reaction feedback(s) on sample strength, we performed rock deformation experiments on "wet" assemblages of mafic compositions in a Griggs-type solid-medium deformation apparatus. Shear strain was applied at constant strain rate ($10^{-5}$ s$^{-1}$) and constant confining pressure (1 GPa) with temperatures ranging from 800 to 900 °C. At low shear strain, the
assemblages that react faster are significantly weaker than the ones that react more slowly, demonstrating that reaction progress has a first-order control on rock strength. With increasing strain, we document two contrasting microstructural scenarios: (1) the development of a single through-going high-strain-zone of well-mixed, fine-grained aggregates, associated with a significant weakening after peak stress and (2) the development of partially connected, nearly monomineralic shear bands without major weakening. The lack of weakening is caused by the absence of interconnected well-mixed aggregates of fine-
grained reaction products. The nature of the reaction products, and hence the intensity of the mechanical weakening, is controlled by the microstructures of the reaction products to a large extent, e.g., the amount of amphibole and the phase distribution of reaction products. The samples with the largest amount of amphibole exhibit a larger grain size and show less weakening. In addition to their implications for the deformation of natural shear zones, our findings demonstrate that the feedback between deformation and mineral reactions can lead to large differences in mechanical strength, even at relatively
small initial differences in mineral composition.





## 1 Introduction

Mafic rocks constitute a large part of the oceanic crust and may be one of the main components of the lower continental crust (Rudnick and Fountain, 1995). The major constituents of mafic rocks, i.e., pyroxene and plagioclase, are

mechanically strong minerals that show crystal plastic deformation only at high temperatures in natural systems (e.g., Rutter and Brodie, 1985; Brodie and Rutter, 1985, 1992). However, there are abundant examples of strongly deformed mafic rocks, even at relatively low temperatures, but invariably these rocks show metamorphic retrogression (Rutter and Brodie, 1985). For instance, concomitant deformation and metamorphism are observed along oceanic detachments, where deep levels of the oceanic mafic crust are exhumed (Harigane et al., 2008; Schroeder and John, 2004). Strongly sheared meta-gabbros are also

present in exhumed subduction belts (Imon et al., 2004; Shelley, 1994; Soret et al., 2019) or in large-scale transcurrent shear zones (Jolivet and Miyashita, 1985; Shelley, 1994). In most cases, the decreasing temperature conditions during deformation result in coeval mineral reactions, often causing strong grain size reduction. The coupling between deformation and reaction is therefore essential to understand the process of strain localization, as observed in mafic mylonites (Brodie et al., 1992).

The pyroxenes typically deform by crystal plasticity at high temperatures and high stresses (e.g., Borg and Handin,

1966; Coe, 1970; Coe and Kirby, 1975; Bystricky and Mackwell, 2001; Bystricky et al., 2016). Mechanical data from high-temperature deformation experiments of mafic rocks are relatively scarce (e.g., Dimanov et al., 2003, 2007; Dimanov and Dresen, 2005; Marti et al., 2017, 2018; Mansard et al., 2020), but the existing studies indicate that weakening processes are dependent on the "deformation history". The study of mylonitic deformation of natural mafic rocks provides insights, at small scale, into the deformation mechanisms and strain localization processes and, at large-scale, into the strength of the lower crust

(e.g., Rutter and Brodie, 1992; Kanagawa et al., 2008). There are two main mechanisms of rock deformation in the viscous deformation regime: (1) Dislocation creep (crystal plasticity; e.g., Paterson, 2013), and (2) Diffusion creep. The latter includes grain-scale diffusion creep, where diffusive mass transfer either occurs through the volume or phase boundaries of individual grains (e.g., Wheeler, 1992) and is the main strain producing process, and diffusion accommodated grain boundary sliding (GBS), where diffusive mass transfer adjusts grain shapes and asperities during cohesive, frictionless sliding (e.g., Ashby and

Verrall, 1973; Paterson, 1991; Langdon, 2006; and references therein). If a fluid is involved and material is dissolved and re-precipitated, the process is often referred to as dissolution-precipitation creep (DPC). Grain boundary sliding and diffusive mass transfer are both always involved, so that the term diffusion creep or DPC are collective terms involving GBS and diffusive mass transfer. Diffusion creep is a grain-size sensitive (GSS) deformation mechanism and may operate at low or high temperature, as well as typically at low stresses (e.g., Paterson, 1995).

Many workers have pointed out the close relationship between strain or reaction-dependent grain size reduction and the activation of GSS creep in a variety of mafic assemblages (e.g., Kruse and Stünitz, 1999; Kenkmann and Dresen, 2002; Baratoux et al., 2005; Kanagawa et al., 2008; Mehl and Hirth, 2008; Menegon et al., 2015; Okudaira et al., 2015). For this reason, grain size reduction is recognized as one of the most significant mechanisms that control rheological properties (e.g., Elyaszadeh et al., 2018; Brodie and Rutter, 1987; Bercovici and Ricard, 2012; Montési, 2013, Platt, 2015). Grain-size-



controlling processes usually include dynamic recrystallization (e.g., Schmid, 1982; Brodie and Rutter, 1987; Behrmann, 1985; Fliervoet and White, 1995; Vissers et al., 1997) and/or metamorphic reactions (e.g., Rubie, 1983; Fitz Gerald and Stünitz, 1993; Stünitz and Fitz Gerald, 1993; Newman et al., 1999), but while dynamic recrystallization is only considered to have a transient mechanical effects (Brodie and Rutter, 1987), a small grain size can be stabilized by new phases in phase mixtures where grain growth is inhibited by the pinning of grain boundaries (e.g., Olgaard and Evans, 1986, 1988; Fliervoet et al., 1997;

Herwegh et al., 2011; Herwegh and Berger, 2004). Furthermore, phase separation and compositional layering commonly form or develop during crystal plastic deformation (dislocation creep) of minerals. During metamorphic reactions and nucleation of new phases, minerals are spatially re-arranged, so that fine-grained mixed-phase zones and polyphase shear bands may develop (e.g., Stünitz and Tullis, 2001; De Ronde et al., 2004, 2005; Kilian et al., 2011; Platt, 2015; Mansard et al., 2018, 2020). Such a spatial re-arrangement controls the bulk strength of the rock, particularly when these phases have a large mechanical contrast.

In particular, the interconnection of weak materials is necessary to induce a significant drop of bulk strength (e.g., Jordan, 1988; Handy, 1994; Dell'Angelo and Tullis, 1996; Holyoke and Tullis, 2006a, b).

    The principal objective of this contribution is to study the effect of initial rock composition on the feedback processes between reaction and deformation. To do so, we have performed rock deformation experiments on "wet" assemblages of plagioclase-pyroxene assemblages in a Griggs-type solid-medium deformation apparatus. As representative of the lower crust,

the starting material was composed of plagioclase (labradorite; plag) and either Mg-rich orthopyroxene (opx) (from peridotite) or Fe-rich opx (from a granulite-facies anorthosite) in order to investigate the effect of different mineral compositions on rock deformation. In this system, the opx deformation properties are assumed to be the same for Mg- and Fe-rich opx. We also performed deformation experiments on amphibole (amph) + plag and pure amph assemblages at similar conditions to extend the study to typical amphibolite facies conditions. All these assemblages were deformed to varying amount of strain, including

at the early stages of deformation. These early stages can be challenging to access when studying natural cases because of successive overprints of deformation stages. In this contribution, we suggest that viscous strain localization is primarily dependent on the ability of minerals to react, and that the feedback between deformation and mineral reactions can lead to large differences in mechanical strength and deformation processes.

## 2 Methods

### 2.1 Experimental procedures

### 2.1.1 Starting material and sample preparation

    We have performed a series of shear deformation experiments in two Griggs-type deformation apparatus at the University of Tromsø (Norway) and at the University of Orléans (France). Experiments were conducted on mineral powders separated from natural materials. Four different starting materials were prepared from different mineral sources: (1) Gem-

quality labradorite ($An_{60}$-$Ab_{38}$-$Or_2$) from Sonora (Mexico) mixed with orthopyroxene ($Wo_1$-$En_{88}$-$Fs_{11}$) from Damaping



peridote (China) and here referred to as Mg-opx; (2) Labradorite ($An_{55}$-$Ab_{44}$-$Or_1$) mixed with orthopyroxene ($Wo_2$-$En_{62}$-$Fs_{36}$) from Hidaka granulite (Japan) and here referred to as Fe-opx; (3) Amphibole (Mg-rich hornblende; composition available in Table 1) from the Massif Central (France) mixed with Sonora labradorite; and (4) Pure Mg-rich hornblende. The pre-separated minerals were crushed in an alumina mortar, then sieved to <100 μm and handpicked, and finally sorted in a distilled water

column to obtain grain sizes between 10–20 μm. Powders were mixed in a 50:50 vol.% ratio in acetone using an ultrasonic stirrer to avoid density/grain size separation (De Ronde et al., 2004, 2005).

To perform experiments, we used a conventional solid-salt, non-coaxial ("general shear") sample assembly with alumina pistons (Précigout et al., 2018). After adding 0.1 wt.% of distilled $H_2O$, the powder was placed between alumina forcing blocks along a 45° pre-cut, so that a shear zone of ~1 mm thick is formed when the deformation experiment starts (Fig.

1). The assembly was wrapped into a nickel foil of 25 μm thick, and then inserted into a weld-sealed platinum jacket. NaCl pieces were used as solid confining medium for both, the inner and outer furnace assembly. The temperature was measured by S-type (Pt/Pt-Rh) thermocouples centered on the sample. Readers are invited to refer to Pec et al. (2012) and Précigout et al. (2018) for further details and descriptions of sample assemblies and experimental protocols employed.

**2.1.2 Experiments and mechanical data processing**

Deformation experiments were conducted at constant shear strain rate of ~2 x $10^{-5}$ $s^{-1}$ to varying amounts of shear strain (see Table 2 for a summary of experimental conditions), at temperatures of 800, 850 and 900 °C, and at confining pressure of 1 GPa. To bring the samples to the desired pressure-temperature (P-T) conditions, both the $\sigma_1$ and $\sigma_3$ pistons are advanced alternatingly between steps of increasing temperature. At the desired P-T conditions, a period of hydrostatic hot-

pressing was applied and the deformation is started by advancing the $\sigma_1$ piston first through the lead piece ("lead run-in") to bring it into contact with the upper forcing block (hit point). During the "lead run-in" stage, the sample is maintained in a more or less isostatic stress state, as the lead protects the sample from becoming deformed. Two series of experiments on Mg-rich opx-bearing assemblages have been performed, one series with a short "run-in" period and a second one with a longer period (Table 2).

At the end of experiments, samples were quenched to 200 °C within 2 to 3 minutes (~150 to 300 °C/min), so that the deformation microstructures and grain size are preserved. Subsequently, the force and confining pressure are decreased simultaneously to room pressure and temperature conditions. During initial stages of the decompression, the differential stress is kept above the confining pressure (~ 100 to 200 MPa) to reduce the formation of unloading cracks.

Experimental data were acquired and recorded using catman® Easy, and then processed after the experiment with a

MATLAB-based program following the "rig" program of Matej Pec (Pec et al., 2016) available at https://sites.google.com/site/jacquesprecigout/telechargements-downloads. The hit point is defined by intersecting the run-in and loading curve tangent lines. After this point, several corrections are then applied to consider the rig distortion and changes



of thickness and surface due to the sample compaction and piston overlap, respectively. The corrected mechanical data are represented in stress vs. strain graphs from the hit-point.


## 2.2 Analytical methods

Double-polished thick sections (~150–200 μm) of the starting materials were prepared for FTIR analysis. Thick sections were prepared from mineral powders for the Mg-rich opx + plag sample and from a natural section of mylonite for the Fe-rich opx + plag sample. Infrared absorption spectra were collected for Mg-opx, Fe-opx and plag using a Nicolet 6700

FTIR instrument at the ISTO (Orléans, France); 128 scans were acquired for each spectrum at a resolution of 4 cm$^{-1}$ with spot size of $40 \times 40$ μm$^2$. Only grain interiors were analyzed by FTIR. The integrated areas of the absorption bands measured between 3750 cm$^{-1}$ and 3000 cm$^{-1}$ were used to calculate the H$_2$O contents using the calibration of Bell et al. (1995) for opx and of Johnson and Rossmann (2003) for plag.

After the experiments, samples were cut parallel to the shear direction and impregnated under vacuum with low-

viscosity epoxy to prepare thin sections. Sample microstructures were analyzed using a scanning electron microscope (SEM – TESCAN MIRA 3 XMU) at ISTO-BRGM (Orléans, France). All SEM analyses have been performed on carbon-coated (20 nm thickness) thin sections at 12-15 KV and a working distance of ~8 mm. Mineral compositions were collected using a CAMECA SX Five electron microprobe analyzer (EPMA) at ISTO-BRGM (Orléans, France). We adopted the following analytical condition: an acceleration voltage of 12–15 kV, a beam current of ~6 nA and a beam diameter of ~1μm.

Thin sections were additionally polished with colloidal silica, and then coated with a thin carbon coat of ~2 nm thick for Electron backscatter diffraction (EBSD) analysis. The EBSD analyses were carried out using an EDAX PEGASUS EDS/EBSD system and the OIM DC 6.4 software (manufacturer EDAX, Mahwah—USA) at ISTO-BRGM (Orléans, France). The operating conditions involved an accelerating voltage of 20–25 kV and a working distance of 15–18 mm. Post-acquisition treatments, which include plotting equal-area lower-hemisphere pole figures of amphibole lattice preferred orientation (LPO),

were performed using the open-source MTEX toolbox for Matlab. Texture strength is expressed through the J-index and M-index (Bunge, 1982; Skemer et al., 2005).

## 2.3 Microstructural analysis

The SEM back scattered electron (BSE) images were used to produce manually digitized grain maps. Grain sizes

were measured from these grain maps by using the public domain software ImageJ (http://rsb.info.nih.gov/ij/). By extracting the area equivalent diameter from these maps, the grain size is defined as the diameter of equivalent circular diameter (d$_{equ}$ = $2 \times \sqrt{\frac{area}{\pi}}$).





Amphibole phase boundaries were traced manually on SEM/BSE images. In the case where grain boundaries are

indistinguishable within amphibole aggregates, EBSD maps have been processed with MTEX to determine the grain size.

## 3 Results

### 3.1 Mechanical data

Depending on the starting material, the mechanical data differ significantly with respect to each other (Fig. 2). At 850

and 900 °C, the Mg-rich opx + plag samples (Fig. 2b-c) are characterized by a pronounced peak stress at shear strains of less

than γ ~1, whereas the Fe-rich opx + plag ones do not show a pronounced peak stress, but a steady-state flow after yield or, at

900 °C, a constant strain hardening behavior (Fig. 2d). At 800 °C, the peak stresses of all opx + plag samples are above the

Goetze criterion ($\Delta\sigma \leq P_{conf}$), which provides an empirical upper limit for viscous creep. Above this limit, samples are expected

to deform by brittle mechanisms (Kohlstedt et al., 1995). The abrupt stress drop of Fe-rich opx + plag samples at 800 °C and

short run-in samples at 800 and 850 °C indicates that slip has occurred at the interface between the sample and one forcing

block (Fig. 2b, d; 557NM, 559NM, and 538NM). The slip is confirmed by sample microstructures and suggests that peak

stress might have been higher without the slip event. In contrast, two Mg-rich opx + plag samples at 900 °C show a pronounced

strain weakening after peak stress. In one case, the sample falls substantially below the Goetze criterion as a result of slip along

the forcing block interface and stabilizes around 800 MPa (OR49NM). In the other case, the sample weakens continuously

after peak stress and reaches almost steady-state for γ ~6.5 (OR41NM). In the experiments with longer run-in periods, the Mg-

rich opx + plag assemblages weaken systematically after peak stress. They reach stresses lower than 400 MPa at 850 °C and

900 °C, corresponding to ~64% (OR38NM) and ~78% (OR34NM) of the peak stress before reaching a quasi-steady-state shear

stress near γ ~6.5 for OR34NM) and γ ~7.7 for OR38NM; Fig. 2c). At 850 °C, the Fe-rich opx + plag sample behavior is no

longer comparable to those of the mixed Mg-rich assemblages, as no weakening occurs after peak stress and a quasi-steady-

state shear stress is reached at low shear strain (γ ~2). At 900 °C, the sample remains weak, but hardens continuously until γ

~5, unlike the other experiments (Fig. 2d).

Compared to the opx + plag assemblages, the amph + plag and pure amph assemblages do not reach the same strength,

regardless of the deformation temperature (Fig. 2e). At 800 °C, the amph + plag assemblage reaches a peak stress of ~563 MPa,

and then slightly weakens to ~483 MPa at γ ~5.8. The pure amph assemblages deformed at 800 and 900 °C show stress-strain

curves with a significant weakening after peak stress at γ ~0.5 to 1.0. However, the sample at 800 °C documents a peak stress

of ~450 MPa higher than the sample deformed at 900 °C.

Our set of experiments reveals two distinct types of mechanical behavior: one that shows a pronounced weakening

after high peak stress (Mg-rich opx + plag and pure amph assemblages) and one without weakening (Fe-rich opx + plag and

amph + plag assemblages) or even hardening (Fe-rich sample at 900 °C, 532NM; Fig. 2d), but deform at considerably lower

stresses (all Fe-rich samples; Fig. 2d).



## 3.2 Mineral reactions and microstructures

Mineral reactions occur pervasively in deformed portions of the samples - there is a clear correlation between strain and reaction progress (cf. Mansard et al., 2020). The pervasive occurrence of mineral reactions induces substantial changes in grain size and spatial distribution of phases. This is particularly prominent in the development of shear bands within the deformed assemblages. For the sake of clarity, the term "bulk shear zone" refers to the whole sample deformed between the two alumina forcing blocks, while the term "shear bands" refers to a localized zone of variable thickness of high shear strain accommodation within the bulk shear zone. In addition, the term "high-strain zone" is also used to refer to domain of coalescence of fine-grained shear bands that are connected as a more or less single zone through the bulk shear zone.

The Mg-rich opx + plag assemblages deformed to high shear strain are characterized by the development of low-strain zones and a high-strain zone in the center of the bulk shear zone (Fig. 3a). Mineral reactions are localized within this single connected zone that traverses the sample from one interface of the forcing block to the other (at 850 °C; Fig. 3a) as fine-grained mixed zones and C-geometry shear bands subparallel to the shear plane (or forcing block interface; Fig. 4a-to-c, 5a-to-f). The C-type shear bands are mainly composed of fine $opx_2$, $plag_2$ and amph that are present as equant grains where their identification is possible by EDS analysis in the SEM (Fig. 4). The original large grains of $plag_1$ and $opx_1$ form porphyroclasts embedded in a mixture of reaction products (Fig. 3a). In contrast to the high strain zone, the reaction products in the low-strain zones usually occur as coronas or rims surrounding $opx_1$ clasts (Fig. 4d-e). The main difference between the assemblages deformed at 850 °C and 900 °C is the degree of strain localization. At 850 °C, strain is highly localized in a ∼250–300 μm wide single zone, while strain is more distributed throughout the sample at 900 °C, and therefore local strain appears lower. Otherwise, the microstructural features are similar to those of the high strain zone (Fig. 4f-g).

At the scale of the bulk shear zone, both the Fe-rich opx + plag and amph + plag assemblages are banded and show a locally developed, non-connected mylonitic foliation characterized by the development of amph-rich shear bands and tails at $opx_1$-porphyroclasts (Fig. 3b-c). These bands wrap around the original and large $opx_1$-porphyroclasts in Fe-rich opx + plag assemblages (Fig. 3b) and in amph + plag assemblages (Fig. 3c). The reaction products in shear bands alternate with aggregates of original grains. The mylonitic foliation is better defined in the amph + plag assemblage because of more pervasive deformation and strain localization (Fig. 3c). The shear bands are more heterogeneously distributed in the Fe-rich opx + plag assemblage deformed at 850 °C (Fig. 3b). This results in the development of an anastomosing network of amph-rich shear bands that are less connected (Fig. 3c), without forming any through-going high-strain zone (Fig. 3a).

The reaction products in the Fe-rich opx + plag assemblages contain less $opx_2$ than the Mg-rich opx + plag assemblages, and amph + $plag_2$ are the main reaction products (Fig. 5g-to-j, 6). From site to site in the Fe-rich opx + plag sample deformed at 850 °C, a variable proportion of opx and plag has reacted. Mineral reactions preferentially occur in strongly deformed areas and form only locally interconnected separate aggregates and shear bands consisting of amph + $opx_2$/cpx or plag (Fig. 6a-b), while in less deformed areas, the reaction products typically occur as thick rims at grain boundaries of extensional sites of $opx_1$ and $plag_1$ clasts (Fig. 6c). The amph grains are elongated and typically form tails extending from $opx_1$





porphyroclasts (Fig. 6b, c). At 900 °C, the Fe-rich opx + plag sample is more homogeneously deformed with a similar

distribution of reaction products as observed at 850 °C (Fig. 6d-e).

The amph + plag assemblage (Fig. 5k-l, 7a-to-c) deformed at 800 °C produced a large amount of reaction products composed of $amph_2$, $plag_2$, clinopyroxene (cpx), and minor zoisite (zo). There is a spatial relationship between amph and cpx, the latter occurring as small grains, predominantly around amph porphyroclasts ($amph_1$) mixed with small grains of new $amph_2$ (Fig. 7a-b). Unlike Fe-rich opx + plag assemblages, the shear bands involve several phases (amph and cpx) in the amph + plag

assemblages (Fig. 5k-l). Regarding the pure amph assemblages deformed at 800 and 900 °C (Fig. 5m-n, 7d-e), mineral reactions are homogeneously distributed with the formation of $amph_2$, cpx and minor zo, quartz (qtz) and garnet (grt). Reaction induces the development of mixture zones of $amph_2$ and cpx and unmixed zones of grt, zo and qtz (Fig. 5m-n).

### 3.3 Chemical composition

The chemical composition of new grains is systematically different compared to that of the original ones. In each assemblage, the original $plag_1$ composition is An55-60. These grains are rimmed by more albite-rich $plag_2$ (An38-45 in Fe-rich opx + plag samples, An 48-55 in Mg-rich opx + plag samples; Figs. 4, 5, 7, 8a-a'). The fine grains in mixed zones are also more albite-rich than the starting $plag_1$ (Fig. 8a-a'). For the $opx_2$, the ferrosilite content increases with respect to the original $opx_1$, regardless of the original $opx_1$ composition (Fig. 8b). The XMg ratio ranges between ~0.85 and ~0.89 in the Mg-rich

$opx_2$ and between ~0.64 and ~0.68 in the Fe-rich $opx_2$ (with $XMg = Mg/(Mg+Fe^{2+})$). The newly formed cpx in the amph + plag and pure amph assemblages has an augite composition (Fig. 8b). It is worth mentioning that the composition of the main reaction products (orthopyroxene and plagioclase) in the Fe-rich opx + plag assemblages are farther away from the starting composition compared to those of the Mg-rich opx + plag assemblages.

The composition of the original amph used in both amph + plag and pure amph experiments is not constant and varies

in the magnesio-hornblende field (Fig. 8c). The newly formed amph for all assemblages have a composition ranging from magnesio-hornblende to tschermakite (Fig. 8c). There are nonetheless four chemically distinct populations of new amph, depending on the composition of the starting mixture, as shown in the plot of Si vs. XMg (Fig. 8c).

### 3.4 OH-content and thermodynamic modeling

Fe-rich opx and plag grains from Hidaka granulite show a broad and asymmetric IR water absorption, with a maximum amplitude at ~3.580 – 3.590 $cm^{-1}$ (Fig. 9). The average $H_2O$-contents (ppm $H_2O$ by weight), calculated between 3750 and 3000 $cm^{-1}$, are $451 \pm 35$ ppm for Fe-rich opx and $226 \pm 24$ ppm for plag. In contrast, there are no $H_2O$ IR absorption bands for Mg-rich opx grains from Damaping peridotite and plag grains from Sonora (Fig. 9).

Pseudosections for the Mg-rich opx + plag and Fe-rich opx + plag assemblages were calculated for different $H_2O$

contents in the simplified but representative system of our experiments ($Na_2O$-CaO-FeO-MgO-$Al_2O_3$-$SiO_2$-$H_2O$) using the



Perple_X 6.6.8 package (Connolly, 2009) combined with the updated database of Holland and Powell. (1998) and the following solution model: amphibole (Dale et al., 2005), orthopyroxene (Powell and Holland, 1999), garnet, clinopyroxene (Holland and Powell, 1998) and feldspar (Newton et al., 1980).

At the experimental P-T conditions, plag, opx, amph, qtz, cpx and grt are expected to be stable phases in the Fe-rich
opx + plag assemblage and plag, opx, amph and qtz in the Mg-rich opx + plag assemblage. The modeling of the Fe-rich opx + plag assemblage agrees rather well with the observed reaction product assemblage, whereas the modeling of the Mg-rich opx + plag assemblage does not match the observed reaction product assemblage.

### 3.5 Initial shear localization

In the Mg-rich opx assemblages, shear deformation is initially localized between the boundaries of original $opx_1$ and $plag_1$, where the nucleation of fine-grained tails of mixed phases that define the general shear foliation occurs (Fig. 10a). Everywhere, the new grains of $opx_2$ are pervasively mixed with $plag_2$ and amph grains (Fig. 10a). In contrast, the initial strain localization in the Fe-rich opx + plag assemblages forms predominantly amph + less $opx_2$ shear bands or tails extending from $opx_1$ porphyroclasts. Amph + $opx_2$ nucleate at boundaries between $opx_1$ and plag, preferentially at extensional ends of $opx_1$
grains, defining σ-type tails that stay connected to the original $opx_1$ grains (Figs. 6b, 10b). In the samples of amph + plag starting material, the reaction products cpx + $amph_2$ + $plag_2$ + zo nucleate at the boundaries between original $Amph_1$ and $plag_1$ grains (Figs. 8a-c, 10c). $Amph_2$ + cpx tend to form mixed layers that surround and stay connected to $amph_1$ porphyroclasts (Figs. 7a, b, 10c), while $plag_2$ + zo form separate aggregates that tend to surround the $amp_2$ + cpx aggregates (Fig. 8c, 10c). Both types of reaction products organize into an anastomosing network of thin $amph_2$ + cpx and $plag_2$ + zo where layers
become progressively more parallel to the shear plane (Fig. 10c).

To summarize, two types of reaction products and related microstructures form in the two different starting material assemblages: (1) In the Mg-rich opx + plag samples, fine-grained phase mixtures are produced by nucleation of reaction products and localize into more or less contiguous bands approximately parallel to the forcing block interface, i.e., the shear plane. (2) In the Fe-rich opx + plag and amph + plag samples, predominantly amph + $opx_2$ and $plag_2$ + zo shear bands and tails
develop from $opx_1$-porphyroclasts and stay connected with these. While fine-grained reaction products in case (1) produce C-type shear bands, the sample fabrics of case (2) develop S-C'-type shear band geometries, the C' bands of which are formed by predominantly amph and $plag_2$.

### 3.6 Abundance and grain size of reaction products

The transition from low-strain to high-strain zones in the Mg-rich opx + plag assemblages is accompanied by a significant grain size reduction (mode of the distribution as the dominant grain size) from ~15 μm to ~0.2 μm (Fig. 11a). The high-strain zones concentrate most of the reconstituted material with roughly ~65% and locally more than ~80% of reaction



products. This proportion is substantially higher compared to that in the low-strain zones (~25%), where no mixed phase layers develop. Similarly, in the mixed phase reacted zones of the original pure amph assemblages, the new grains are very

small (~0.6 μm) and they represent nearly ~61% of the assemblage (Fig. 11b). The amph grain size is also reduced in Fe-rich opx + plag assemblages compared to the starting material (Fig. 11c), but the reaction products are not as pervasively mixed and show a grain size that is approximately one order of magnitude larger (1.9 μm) than that in the Mg-rich opx + plag samples (0.2 μm; Fig. 11a). In the 850 °C sample, the reaction products are largely connected to $opx_1$ porphyroclasts, and their proportion increases with increasing proximity to higher-strained portions of the shear zone (~26%). In these Fe-rich opx +

plag assemblages, amph is the main reaction product (~17%). In contrast, the amount of reaction products at 900 °C remains roughly constant at approximately ~29% and there is no strain gradient. Finally, in the amph + plag starting material, the nucleation in amph-rich shear bands is more extensive (~36%) and the grain size of cpx reaction products decreases to ~1.3 μm (Fig. 11d).

**3.7 Analysis of SPO and amphibole LPO**

Amph formed during deformation of Fe-rich opx + plag assemblages exhibits a distinct SPO oriented at ~30° to the shear direction (Fig. 12a-b). These amph aggregates occur as reaction rims around original $opx_1$ grains and form partly interconnected aggregates that define S-C´fabrics. $Plag_2$ and $opx_2$ also reveal a well-defined SPO similarly oriented to amph. The preferred orientation of amph forms an angle of ~30° with the shear plane, slightly greater than that of plagioclase and

original opx (Fig. 12a-b). In addition, the amph reaction seams in high-stress sites of opx porphyroclasts is significantly thinner compared to that in low-stress or extensional sites, indicating that amph grows preferentially in strain shadows. In the higher strained portions of the shear zone deformed at 850 °C, the foliation defined by amph-dominated layers rotates and is now well oriented sub-parallel to the boundaries of C´ shear bands or the shear plane (Fig. 12c). Within these shear bands, amph grains show a moderate LPO with [001] axes aligned subparallel to the boundaries of C' shear bands and poles (100) normal

to the shear plane (Fig. 12c-d). In the mixed zones of Mg-rich opx + plag samples, the fine grains are characterized by equant grain shapes with an aspect ratio of 1.23 (Fig. 13), and a weak preferred orientation either parallel or at 45° to the shear plane (Fig. 13).

**4 Discussion**

**4.1 Nucleation, grain size reduction and phase mixing**

In this study, the deformation of two-phase assemblages is accompanied by the nucleation of new grains with 1) systematic difference in composition between new grains and parent grains (Fig. 8), 2) significant grain size reduction (Fig. 11) and 3) a new spatial arrangement of reaction products into fine-grained, more or less mixed phase zones (Fig. 5). Based



on these observations, we infer that grain size reduction and phase mixing both result from the nucleation of new phases (e.g., Kruse and Stünitz, 1999; Kenkmann and Dresen, 2002; Kilian et al., 2011; Linckens et al., 2015; Platt, 2015; Précigout and
Stünitz, 2016; Mansard et al., 2018). Despite common features, our samples have developed two sets of microstructures, distinguished mainly by the size and spatial arrangement of reaction products and the degree of phase mixing. We discuss these differences below.

In the Mg-rich opx + plag assemblages, a strain gradient is clearly observed and expressed by the development of low- and high-strain zones (Fig. 3a, 4). This is particularly notable in the sample deformed at 850 °C. The transition from low-
to high-strain zone is accompanied by drastic grain size reduction and increase in reaction products within localized mixed zones by a factor of more than 2 (Fig. 11a). The grain size of reaction products in the mixed high-strain zones (~0.2 μm) is about two orders of magnitude smaller than the opx grain size of the starting material and the low strain regions (~10-20 μm; Fig. 11a). Similarly, an extensive nucleation of reaction products within mixed-phase zones is documented in the pure amph starting material (Fig. 7d-e). The reaction products are also very small, far below one micron (~0.6 μm; Fig. 11b). As
documented in Mansard et al. (2020), such a correlation between deformation, mineral reactions and related grain size reduction in Mg-rich assemblages has been also observed in experimentally deformed fine-grained gneiss (Holyoke and Tullis, 2006 a, b), plagioclase + olivine samples (De Ronde et al., 2005; 2007), and plagioclase + pyroxene samples (Marti et al., 2018).

In the Fe-rich opx + plag assemblages, amph is by far the more abundant product, and there is less phase mixing (Fig.
5g-j). The reaction products consist mainly of thick rims around the opx$_1$ (Fig. 6c) or as amph-dominated tails and shear bands (Fig. 6a-b). The abundance of reaction products increases with increasing proximity to strongly deformed portions of the shear zone. The reaction products have a grain size about one order of magnitude smaller than the starting material (~1.9 μm), but one order of magnitude larger than the mixed reaction products in the Mg-rich opx + plag assemblages (Fig. 11). Similarly to the Fe-rich opx + plag  assemblages, amph-rich shear bands have developed in the amph + plag assemblages (Fig. 7a-c), and
the grain size, reduced by about one order of magnitude compared to the starting material, is one order of magnitude larger than in the mixed zones in the Mg-rich opx + plag assemblages (Fig. 11).

Two types of major microstructures can be distinguished: (1) in one case, intense grain size reduction by 2 orders of magnitude is produced by nucleation of reaction products in pervasive and layered phase mixtures (Mg-rich opx + plag and pure amph starting materials); (2) in the other case, the reaction products are less pervasively mixed and develop aggregates
that are dominated by amph extending from opx-porphyroclasts. These aggregates have a grain size of one order of magnitude larger than (1) and may form C´ shear bands or tails connected to opx porphyroclasts.

## 4.2 Deformation processes

As the reaction products in Mg-rich opx + plag samples occur in pervasively mixed high-strain zones and layers with equant grains of a size far below one micron, the dominant deformation mechanism is inferred to be one of grain size sensitive
creep. The absence of well-developed layering of separate phases and absence of strong elongation of individual grains during



GSS creep suggests that grain boundary sliding makes the dominant kinematic contribution to the finite strain ("Rachinger sliding"; e.g., Langdon, 2006). Such a deformation mechanism should probably be termed "diffusion-accommodated grain boundary sliding" (GBS) or, more general, as a fluid has been present and solution precipitation is the probable transfer mechanism, "dissolution precipitation creep" (DPC). The pronounced weakening of the samples in combination with localization of strain made strain rate stepping tests problematic in our samples, so that the deformation mechanism is primarily identified based on microstructures. At first, the microstructures are similar to other cases where such deformation mechanisms have been identified (e.g., Marti et al., 2017, 2018; De Ronde et al., 2005; Stünitz and Tullis, 2001; Holyoke and Tullis, 2006a, b; Tasaka et al., 2016, 2017; Getsinger and Hirth, 2014). The transition from low- to high-strain zones in the Mg-rich opx + plag assemblages marks the transition from a two-phase aggregate with strong phases (starting material) potentially deforming by dislocation creep to a material deforming by grain size sensitive mechanisms, including DPC and/or GBS (e.g., Boullier and Gueguen, 1975; Kerrich et al., 1980; Schmid, 1982; Brodie and Rutter, 1987; Kilian et al., 2011), where samples deform by low bulk stresses (Fig. 2).

The spatial arrangement of mineral phases in mixed aggregates by nucleation impedes grain growth (e.g., Olgaard and Evans, 1986, 1988). As documented by Mansard et al. (2020), the mixing of mineral phases is homogeneous and starts from peak stress, in favor of GBS accommodated by DPC (e.g., Fliervoet et al., 1997; Kilian et al., 2011; Linckens et al., 2011; 2015). The original $plag_1$ almost completely disappeared in the high-strain zones of Mg-rich samples, while the original $opx_1$ remain as small clasts embedded in fine-grained mixed zones (Figs. 3a, 4a-to-c). Although the $opx_1$ is still present, its proportion, aspect ratio and size decrease compared to that in the low-strain zones (Fig. 4). These microstructures suggest that $opx_1$ grains act as rigid particles affected by dissolution and together with $plag_1$, represent a source of elements required for the development of the ductile fine-grained zone deformed by diffusion-accommodated GBS.

In the Fe-rich opx + plag samples, the deformation mechanism is also assumed to be predominantly DPC, as demonstrated by the fact that (1) the samples deform at lower stresses well below the Goetze criterion, and (2) the microstructures that consist of elongate mineral aggregates of reaction products dominantly grow in the extension direction, shear bands or in dilatant sites (Fig. 2, 6, 12). The nucleation of reaction products, anisotropic growth and local dilatancy could also require the operation of GBS ("Lifshitz sliding"; Paterson, 1990; Langdon, 2006). The grain sizes of the reaction products in the Fe-rich opx + plag samples are larger, resulting in higher flow stresses compared to the final stresses – after weakening – of the Mg-rich opx + plag samples (Fig. 2, 11; e.g., Mansard et al., 2020).

In addition, the amphibole fabric displays a fairly moderate, but consistent LPO with [001] axes aligned sub-parallel to the boundaries of C' shear bands (Fig. 12). This type of LPO is typical for naturally deformed amphibole (e.g., Berger and Stünitz, 1996). It is generally accepted that significant LPO of minerals is attributed to viscous deformation dominated by dislocation creep (e.g., Nicolas and Christensen, 1987; Knipe, 1989; Wenk and Christie, 1991). However, an increasing number of studies have found that LPOs can develop without the dominance of dislocation creep, as shown in olivine (e.g., Sundberg and Cooper, 2008; Miyazaki et al., 2013, Précigout and Hirth, 2014), plagioclase (e.g., Barreiro et al., 2007) and amphibole (e.g., Getsinger and Hirth, 2014). The amph LPOs presented here are similar to the ones documented in experimentally





deformed amphibolite by Getsinger and Hirth, (2014), and to those of natural samples deformed at lower crustal conditions (e.g., Berger and Stünitz, 1996; Getsinger et al., 2013; Menegon et al., 2015; Okudaira et al., 2015). These studies have demonstrated that oriented grain growth of pyroxenes and amphibole can lead to consistent amphibole/pyroxene LPO, even though diffusion creep is the dominant deformation mechanism. We thus interpret the presence of amphibole LPO in our samples as resulting from an oriented growth with the fastest growth direction (c-axis) in the flow direction. This supports an

important contribution of metamorphic re-equilibration, chemical transport, and hence, DPC in the development of amph mylonitic foliation in the Fe-rich and amph + plag assemblages (e.g., Bons and Den Brok, 2000; Berger and Stünitz, 1996).

### 4.3 Formation of polyphase vs. monophase shear bands and implications for the degree of rheological weakening

#### 4.3.1 Polyphase shear bands

In Mg-rich opx + plag assemblages at 850 °C, the mixture zones are strongly connected to form a single high-strain zone that traverses the sample through the center from one interface of the forcing block to the other (Fig. 3). The deformation microstructures at early stages of our experiments suggest that phase mixing occurs at phase boundaries with the nucleation of small equant grains in polyphase aggregates and layers (Figs. 10a). These polyphase aggregates, into which strain is partitioned, are composed of $opx_2$, $plag_2$ and amph. It has been documented by many authors (e.g., Kruse and Stünitz, 1999; Kenkmann

and Dresen, 2002; Linckens et al. 2015) that phase mixing can be produced by the nucleation of new phases within clast tails. The effect of mixing on grain size is significant as grain growth is impeded by the nucleation of second phases and preserves a small grain size in the mixture zones (Fig. 11a), producing stable microstructures (e.g., Olgaard and Evans, 1986; 1988; Olgaard, 1990; Stünitz and Fitz Gerald, 1993; Herwegh and Berger, 2004; Warren and Hirth, 2006; Farla et al., 2013). The opening of cavities or formation of dilatant sites in which material can nucleate (e.g., Fusseis et al., 2009; Platt, 2015; Menegon

et al., 2015; Précigout and Stünitz, 2016; Précigout et al., 2017; Gilgannon et al., 2017) may be an additional factor to promote mixing and leading to stabilize small grain sizes.

#### 4.3.2 Amphibole-dominated σ-tails and shear bands

    In the Fe-rich opx + plag samples, the majority of reaction products occurs in strongly deformed portions of the shear

zones, indicating a close relationship between deformation and mineral reactions (Fig. 3b). These portions are not characterized by the development of intensely mixture zones with C-type mylonite geometries, as in the Mg-rich opx + plag assemblages. Instead, deformation and reaction have induced the formation of shear bands or, more frequently, σ-tails at the tip of elongated amph or amph + $opx_2$ (Fig. 5, 6). These tails stay connected with the original porphyroclasts, forming S-C or S-C´-fabrics. Similarly, the amph + plag assemblage is banded and shows a sub-horizontal mylonitic foliation characterized by the





development of amph-rich shear bands (Fig. 3c). These shear bands have a similar geometry as those of the amph + opx$_2$ bands
        and σ-clasts of the Fe-rich opx + plag assemblages.

### 4.3.3 Effects of shear band interconnection on the degree of rheological weakening

            Shear bands connect differently depending on the composition of the starting material. In the Mg-rich opx + plag
samples, strain tends to produce mixed and connected fine-grained bands with C-type geometry in high-strain zones (Fig. 3a,
        4, 10a, 11a). This gave rise to the development of a single through-going high-strain zone that probably contributed to strongly
        weaken the samples after peak stress (Fig. 2, 3a). And the samples are so weak that we consider the fine-grained aggregate of
        reaction products as almost sustaining the whole sample strength. In contrast, the Fe-rich opx + plag and amph + plag mixtures
        tend to form clusters and only locally connected amph-rich σ-tails at porphyroclasts (S-C- or S-C'- type geometries; Fig. 3b-
c, 5g-l, 6 and 10b). When the shear bands and tails of clasts are only partially or non-connected, there is no peak stress, and
        hence, no subsequent weakening (Fig. 3b; Fe-rich opx + plag assemblages). This feature may be caused by two aspects: (1)
        the absence of highly connected aggregates of reaction products; and (2) the fact that reaction products have a larger grain size
        and are less intensely mixed, so that GSS creep occurs at lower strain rate. The connectivity of the reaction products appears
        to be also affected by their geometry; the fine-grained C-type bands are parallel to the shear plane and seem to connect much
more easily than the local S-C- and S-C'-type tails and shear bands.

            The fact that the connectivity of weak zones has a major effect on the bulk sample strength has been documented by,
        e.g., Pec et al. (2012, 2016), Palazzin et al., 2018, Richter et al. (2018) and is definitely a major rheological factor in these
        mafic samples. However, in addition to the connectivity, we also documented different microstructures in the weak parts of
        each sample, pointing to a difference in GBS/DPC mechanisms. The less intensely mixed layers, tails and shear bands in the
Fe-rich assemblages display higher aspect ratios of the reaction products, i.e., more fiber-like grains (Fig. 6c; 10b, 12a-b).
        These microstructures constitute a type of DPC that is tending more towards grain-scale DPC, where ideally (in the end-
        member case) the grain shapes would reflect the finite strain and where grain boundary sliding is more a type of Lifshitz-
        sliding. We cannot consider the Fe-rich opx + plag samples as representative of the end-member case, but they approach this
        situation far more than the fine-grained, well mixed mylonitic bands of equant grains in Mg-rich opx + plag samples (Fig. 4a-
b; 13), where DPC is probably dominated by GBS or Rachinger-sliding. The combination of relatively fast strain rates in the
        fine-grained layers and strong connectivity of the reaction product zones causes the pronounced weakening in the Mg-rich opx
        + plag samples, emphasizing the importance of grain size on material strength (Mansard et al., 2020). Such expected higher
        strain rates in aggregates dominated by Rachinger sliding has been pointed out by Paterson, (1990).






### 4.4 OH speciation and concentration in the starting material

Our absorption spectra have been compared with reference spectra available for plagioclase (Johnson and Rossmann, 2003, 2004; Johnson, 2006) and orthopyroxene (Skogby, 2006) in the literature. While IR spectra we obtained in plagioclase and opx are very similar to one another, they strongly differ from the reference spectra, which correspond to structural OH or 450   $H_2O$, i.e. to molecules with specific position and orientation in the structure of the host mineral. The spectra in plagioclase bear the largest similarities with reference measurements of fluid inclusions and alteration products (Johnson and Rossman, 2004), i.e., OH or $H_2O$ molecules not structurally bound to the host solid. This interpretation is also favoured by the very similar shape of the spectra in plag and opx (Fig. 9), whereas structural OH or $H_2O$ in these minerals yield different spectra.

According to this interpretation, the calibration coefficient for the $H_2O$ content should be different from that of the 455   mineral-specific ones we have used. However, using the general coefficient from Paterson (1982) would not significantly affect the estimates of $H_2O$ concentration in the starting material: the integrated absorption coefficient used in Fig. 9 is 80600 L/(mol $H_2O.cm^2$) for opx (Bell et al., 1995), and 107000 L/(mol $H_2O.cm^2$) for plag, i.e. relatively similar to the estimate of 82200 L/(mol.$cm^2$) for quartz and other silicates (Paterson, 1982).

Bearing in mind these limitations, the amount of $H_2O$ initially present in the minerals of the Fe-rich system is of the 460   order of 451 ppm for pyroxene and 226 ppm for plagioclase, whereas it is effectively zero in the Mg-rich-system. The total amount of $H_2O$ in pyroxene and plagioclase in the Fe-rich system is ~677 wt ppm, i.e., more than half with respect to the addition of $H_2O$ to the powder. The initial amount of $H_2O$ estimated by FTIR is a lower bound as only grain interiors were analyzed by FTIR, while crushed polycrystalline aggregates include grain boundary area with additional adsorbed $H_2O$ (Palazzin et al., 2018). In summary, it is estimated that the total amount of $H_2O$ in the Fe-rich system has been ∼50% higher 465   than that in the Mg-rich-system.

### 4.5 Role of H2O availability on reaction

In the Fe-rich opx + plag samples, a large amount of amph is formed with a minor amount of opx2, whereas in the Mg-rich opx + plag samples, the nucleation of a new opx2 is more abundant. The thermodynamic modeling of the Fe-rich opx 470   + plag system has produced a result similar to that of Okudaira et al. (2015) as an extension to higher pressures and temperatures, and rather consistent with the observed phase compositions, although the predicted garnet was not observed, and cpx is of lower abundance than predicted by the model. However, it was not possible to model the observed reactions in the Mg-rich opx + plag system in terms of phase compositions. The reason for the inadequate modeling is most likely the somewhat inadequate activity-composition relation in the Mg-rich system. Due to this issue, it is impossible to calculate differences in 475   free energy between the two systems as a possible explanation for the different reaction kinetics.

Another difference between the two starting materials is the higher content of $H_2O$ in the Fe-rich system. Considering that amphibole contains 2 wt.% of $H_2O$, the amount of $H_2O$ required to form 17% of amphibole would be 3400 wt. ppm,





which is a bit higher than the total amount of H2O present in the sample. Our estimations of 17% amphibole content is also a rough estimate. In any case, it is probable that a large amount, if not all, of the H2O is used up by the amphibole formation,

even though the total amount of H2O in the Fe-rich system is greater to begin with. Conversely, the amount of amphibole formed in the Mg-opx + plag assemblage is subordinate, so that, despite a lower total amount of H2O in the starting material, there probably is free H2O present in the Mg-rich system. It could be speculated that the presence of more H2O in the grain boundary region could lead to a more disperse nucleation of reaction products in the Mg-rich system.

**4.6 Influence of reaction on material strength**

**4.6.1 Behavior at the onset of deformation: peak strength**

Considering that the rheological behavior is strongly controlled by the reaction products, it is inferred that the Fe-rich opx + plag assemblages do not develop a peak stress behavior and initially deform at lower stresses than the Mg-richopx + plag assemblages because in the Fe-rich assemblages reaction products nucleate faster than in the Mg-rich assemblages at early

stages of the experiment (Fig. 2b-d, 14). This is consistent with the recent study of Mansard et al. (2020) showing that for samples where reaction products nucleate at an early stage with respect to the onset of deformation, the peak strength is lowered. The system that reacts faster is the Fe-rich one, which contains a greater concentration of $H_2O$. This water is present inside the grains of the starting material, whereas the added water is located along the grain boundaries. These differences suggest that the presence of $H_2O$ in inclusions or aggregates inside grains may have triggered the onset of hydration reactions,

so that, possibly, the higher initial content in $H_2O$ and its location inside grains in the Fe-rich system may have had a positive effect on the kinetics of reaction. The reactions have commenced earlier in the Fe-rich system, presenting a weakening agent in the early stages of the deformation.

**4.6.2 Behavior at large strain**

During later stages of our experiments, the highest proportion of reaction products and the smallest grain size is documented for the Mg-rich opx + plag samples (Fig. 11, 14), which also record a far more pronounced weakening with respect to the Fe-rich samples (Fig. 2, 14). The phase mixing and fine grain sizes of the reaction products in Mg-rich opx + plag samples cause greater strain partitioning and suggest a faster nucleation rate after peak stress, even though the Fe-rich opx + plag samples react faster at early stages of experiment (Fig. 8, 14). Deformation and reaction products are strongly localized

and connected in high-strain zones of the Mg-rich opx + plag assemblages (Fig. 3a, 10a), whereas the reaction products in the Fe-rich opx + plag samples are poorly or non-connected (Fig. 3b, 10b). In the case where the viscosity of the reaction products is very low (fine-grained mixture zones in the Mg-rich opx + plag) compared to the starting material, the reacting domains tend to connect much better during the deformation (Fig. 14). It has been demonstrated that deformation enhances the kinetics



of mineral reactions (De Ronde and Stünitz, 2007; Mansard et al., 2020), so that the significant localization of deformation in

the Mg-rich opx + plag samples may account for the more advanced reaction progress in those samples. On the contrary, if the reaction products are stiffer – in Fe-rich opx + plag samples, amphibole appears to be one of the strongest silicates (Brodie and Rutter, 1985; Berger and Stünitz, 1996) – or harden during the reaction (see Fe-rich opx + plag at 900 °C; Fig. 2d), the feedback effect of enhancing reaction kinetics appears to be limited and no weakening is observed (Fig. 2). Therefore, the viscosity of the Fe-rich opx + plag and amph + plag assemblages is related to their ability to transform and connect.

The reason why the reaction products in Mg-rich samples characterize by fine-grained layers that do not stay attached to the original $opx_1$ clasts is not clear yet, but this might be related to the difference in the amount and distribution of $H_2O$. Indeed, in the Fe-rich system, the reaction products are dominated by a single-phase material, i.e., amphibole (Fig. 6). In contrast, although the very small grain size did not allow any quantitative estimates of the Mg-rich reaction products, much more phase mixing occurs in these latter. The difference in nature and proportion of reaction products between Fe- and Mg-

rich systems is a major control factor for their later resistance evolution during connectivity and reaction progress. This situation illustrates that substantially more work is needed to understand the relationship between mineral reaction and mechanical behavior.

### 4.7 Geological application

According to the results of this study, the rheological behavior of two-phase mixtures of mafic composition is dependent on mineral reactions, as reactions can trigger strain localization and weakening, even at low shear strain. In two different starting materials, the reactions coupled with dissolution precipitation creep (DPC) in combination with grain boundary sliding allow the mafic assemblage to be deformed. Without mineral reactions, the mafic assemblage of pyroxene and plagioclase would be too strong to be deformed by dislocation creep at the applied strain rates (Mansard et al., 2020). This

situation is the same in naturally deformed rocks: below ~700 °C, pyroxene and plagioclase assemblages do not show significant crystal plastic deformation (e.g., Brodie and Rutter, 1985, 1992). The partly extensive deformation of mafic rocks at lower metamorphic grades appears to be dependent upon mineral reactions, which can induce the development of rheologically weaker phases, or in the formation of fine-grained aggregates deforming by GSS creep, or in both (e.g., Brodie and Rutter, 1985; Brodie et al., 1992; Handy and Stünitz, 2002; Brander et al., 2012; Okudaira et al., 2017).

At low shear strain and identical P-T conditions, the small difference in mineral reactions between the Fe-rich opx + plag and Mg-rich opx + plag assemblages induces clear differences in mechanical strength at peak stress and during subsequent weakening. These observations are crucial as shear zones are known to represent preferential regions for mineral reactions to occur, either owing to deformation (e.g., Brodie, 1980) or to the presence of fluids (e.g., Etheridge et al., 1983) during protracted shearing (e.g., Brodie, 1980; Philippot and Kienast, 1989; Newman et al., 1999). Thus, local spatial variations in

mineral reactions can induce gradients in deformation due to sufficient differences in the mechanical behavior to localize strain (e.g., Holyoke and Tullis, 2006a, b). In addition, our observations suggest that the mechanical strength at peak stress is





controlled by the ability of the material to react and to connect the weak material, and not necessarily by the strength of the reaction products. In fact, the Fe-rich opx + plag assemblages produce intrinsically stronger reaction products (more amphibole) than the fine-grained mixed aggregates and layers in the Mg-rich opx + plag assemblages, and yet Fe-rich opx + 545 plag assemblages are weaker at low shear strain because they react faster. It is speculated that this difference in reaction kinetics is in part controlled by the amount of $H_2O$ present. The higher $H_2O$ content in the Fe-rich opx + plag assemblages may have had a positive effect on the kinetics of reaction. This aspect on the influence of $H_2O$ availability on the strength of materials is important because shear zones have long been recognized as a preferred locus for fluid migration (e.g., Austrheim, 1987; Newton, 1990; Selverstone et al., 1991; Gueydan et al., 2004; Angiboust et al., 2011), particularly due to the development of 550 synkinematic porosity resulting from dilatancy at grain boundaries (Tullis et al., 1996), creep cavitation (Fusseis et al., 2009; Menegon et al., 2015; Précigout and Stünitz, 2016; Précigout et al., 2019) or from dehydration reactions and associated fracturing (Plümper et al., 2017).

Our results suggest that small initial differences in mineral reactions in natural materials can trigger strain localization and lead to large differences in mechanical strength, regardless of the intrinsic strength of the reaction products. These 555 observations mainly apply to the initiation and early stages of shear zone formation, as once the incipient stages of strain localization and weakening are initiated, additional mechanisms are involved and control the development of the shear zones in space and time.

The development of natural shear zones commonly arises into distinctive monophase layers (e.g., Berthé et al., 1979; Gapais, 1989) and/or polyphase mixtures (e.g., Fliervoet et al., 1997; Keller et al., 2004; Warren and Hirth, 2006; Kanagawa 560 et al., 2008; Raimbourg et al., 2008; Okudaira et al., 2015) that may be found in the same shear zone as alternating layers (e.g., Kenkmann and Dresen, 2002; Kilian et al., 2011; Oliot et al., 2014; Mansard et al., 2018). In our case, the progressive deformation of two-phase assemblages also shows the development of (1) strongly localized polyphase high-strain zone associated with a significant weakening after peak stress (Mg-rich assemblages), and (2) only locally connected σ-clast bands that do not induce major weakening (Fe-rich opx + plag and amph + plag assemblages). Strain localization into reaction- 565 derived fine-grained mixture is also well documented in nature with the dominance of diffusion-accommodated GBS and/or DPC mechanisms (e.g., Newman et al., 1999; Handy and Stünitz, 2002; Kanagawa et al., 2008; Okudaira et al., 2015) and is expected to promote major rheological weakening (e.g., Fliervoet et al., 1997; Svahnberg and Piazolo, 2013; Warren and Hirth, 2006; Platt, 2015). The development of amph-rich σ-shaped aggregates and layers, notably through dissolution and precipitation processes, is also commonly documented in mid- to lower-crustal banded mylonites (e.g., Berger and Stünitz, 570 1996; Imon et al., 2002; Imon et al., 2004; Baratoux et al., 2005; Getsinger et al., 2013; Elyaszadeh et al., 2018; Giuntoli et al., 2018).

When applied to natural shear zones, our results suggest that (1) the ability of minerals to react controls the portions of rocks that deform, and (2) minor chemical and mineralogical variations (including the $H_2O$ available for reaction) can considerably modify the strength of deformed assemblages. Then, significant strain localization and partitioning at high shear 575 strain, expressed by the development of interconnected high-strain zones (Mg-rich opx + plag), illustrates that fine-grained





polyphase mixed zones can become weaker than coarse-grained, poorly-connected ones (Fe-rich opx + plag and amph + plag). These results emphasize the importance of strain partitioning, grain size reduction, and interconnectivity of weak material as a primary control of lithospheric strength. This also suggests that the rheology of mafic rocks, which constitutes a large part of the oceanic crust and may be one of the main components of the lower continental crust, cannot be summarized as being

rheologically controlled by monophase materials (e.g., Dimanov and Dresen, 2005).

**5 Conclusions**

Shear deformation experiments of monophase and polyphase assemblages show a clear relationship between deformation and mineral reactions. A greater volume of reaction products is documented with increasing strain. This study shows that the premise of mineral reaction and strain localization has a major impact on sample strength because it conditions

the resistance of the assemblage at low shear strain. Indeed, the Fe-rich opx + plag assemblages deform at lower stresses than the Mg-rich opx + plag assemblages because they react faster at early stages of experiment. Thus, chemically similar assemblages may have significantly different strength development depending on the ability of these minerals to react. It is suggested that the availability of $H_2O$, among other factors such as mineral composition, has a positive effect on the kinetics of reaction. With increasing strain, there are two very contrasted pathways in the material evolution, controlled by the

properties of the reaction: in one case (Fe-opx + plag and amph + plag assemblages), the reaction products (mostly amphibole) have elongated grain shapes, a larger grain size and poor connectivity. As a result, they do not show significant weakening. In such a case, the reaction products are only partially connected, the material deforms by grain scale DPC, and strength of the bulk sample is stable and the feedback effect of deformation on reaction is limited. In the other cases (Mg-rich opx + plag and pure amph assemblages), the fine-grained, intensely mixed reaction products of equant grains weaken considerably during the

experiment and end up much weaker than the initial material. There, mechanically weak zone of reaction product deforms by GBS-dominated DPC and tends to form an interconnected zone that leads to the weakening of the bulk sample, and the drastic increase in proportion of reaction products with strain suggests a large feedback effect. Overall, the value of the initial peak stress (that is responsible for determining where strain will start localizing) or the large weakening associated with the formation of fine-grained products (that determine whether shear bands thicken or not), seem to be mainly controlled by the

chemical reactions.



**Data availability**

Experimental data were processed using a MATLAB-based program inspired from the "rig" program of Dr. Matej Pec (Pec et al., 2016) and available at https://sites.google.com/site/jacquesprecigout/telechargements-downloads.

**Author contribution**

    Nicolas Mansard, Holger Stünitz, Hugues Raimbourg and Jacques Précigout designed the experiments and Nicolas Mansard carried them out. Nicolas Mansard carried out the various analyses after the experiments (e.g., SEM, EPMA). Alexis
Plunder worked on the thermodynamic modeling part while Lucille Nègre contributed to the FTIR data acquisition. Nicolas Mansard prepared the manuscript with contributions from all co-authors.

**Competing interests**

    The authors declare that they have no conflict of interest.

**Acknowledgements**

This work has received funding from (1) the European Research Council (ERC) under the seventh Framework Programme of the European Union (ERC Advanced Grant, grant agreement No 290864, RHEOLITH), (2) the Labex VOLTAIRE (ANR-10-LABX-100-01), and (3) the project FluPrism (CNRS INSU, funding program SYSTER). We gratefully acknowledge the help provided by Patricia Benoist and Ida Di Carlo for analytical support, Sylvain Janiec for the preparation of thin sections and Kai Neufeld for EBSD data acquisition. The authors wish to thank Mark Zimmerman for generously
providing the Damaping peridotite and Laurent Arbaret for the Massif Central amphibolite material.

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





**Figure 1:** Drawing of the sample assembly in the Griggs-type apparatus.











**Figure 2:** Mechanical data. (a) Terms used to describe the different stages of an experiment. (b-c) Differential stress (MPa) versus shear strain (γ) showing the mechanical behavior of the Mg-rich opx + plag assemblages deformed at temperatures ranging from 800 to 900 °C, at constant confining pressure of 1 GPa and strain rate of $10^{-5}$ s$^{-1}$. The difference between (b) and (c) is related to the different duration of the "run-in" section, i.e. time spent at P-T conditions before the hit point. Mechanical data for Fe-rich opx + plag (d), amph + plag and pure amph (e) assemblages are also plotted in stress-strain graphs. opx = orthopyroxene, pl = plagioclase, amph = amphibole.







**Figure 3:** Distribution of reaction products after deformation in the Mg-rich opx + plag (a), Fe-rich opx + plag (b), and amph + plag (c)

assemblages. For each assemblage, a manually digitized overview of the shear zone is associated with a zoomed part of it. The mechanical

data associated to these assemblages are also represented in stress-strain graphs.







**Figure 4:** SEM-BSE images representative of microstructures observed in deformed Mg-rich opx + plag assemblages. $opx_2$ and $plag_2$ are
the main reaction products. (a-b) At 850 °C, mineral reactions are mainly localized in the high-strain zones in the form of fine-grained mixed
zones. (c) The original $plag_1$ almost completely disappears. (d) In low-strain zones, the reaction products appear as coronas around the
original opx1 and as aggregates. (e) $opx_1$ is locally fractured and filled with reaction products. (f-g) Similar microstructures are observed at
900 °C, although deformation is less localized compared to the assemblage deformed at 850 °C. opx = orthopyroxene, pl = plagioclase,
amph = amphibole.









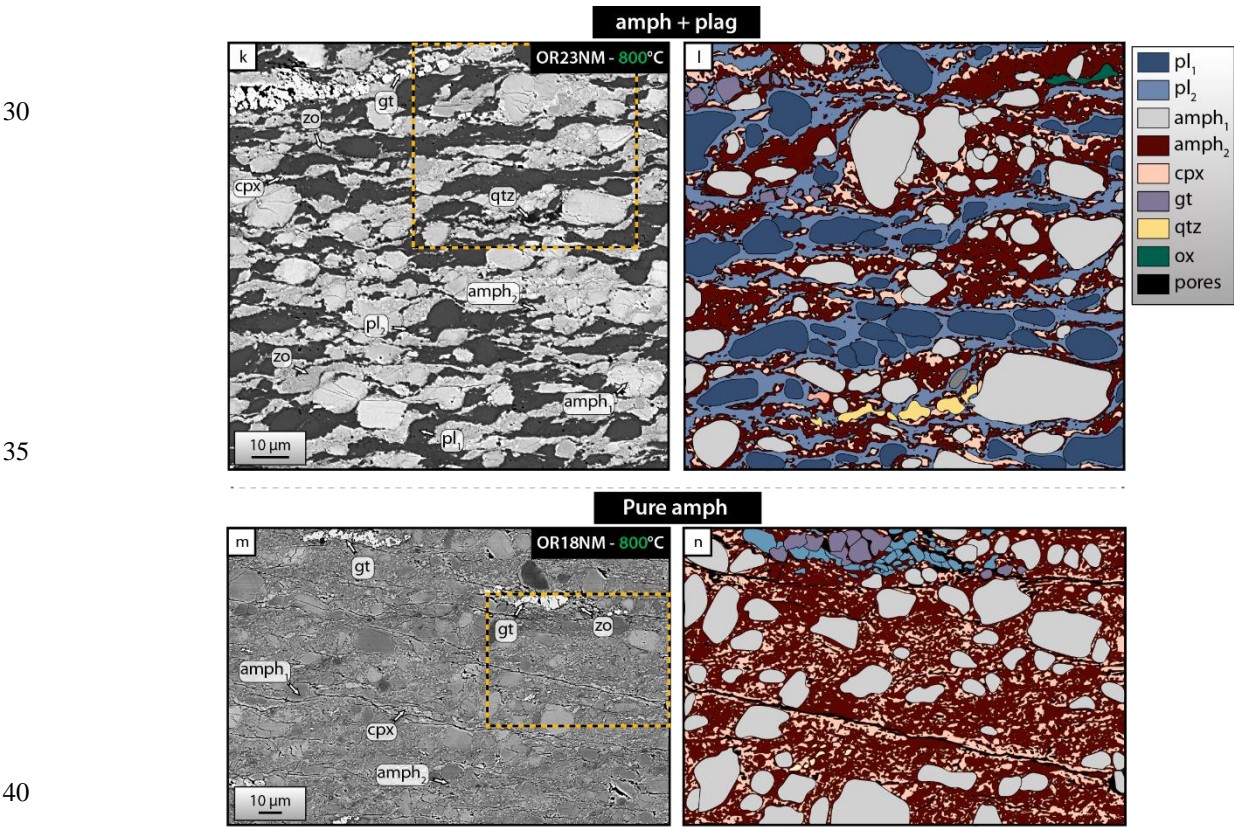

**Figure 5:** SEM-BSE images representative of the different small-scale microstructures encountered in the assemblages and their respective manually digitized phase maps. Please note that the resolution is too low to resolve the full extent of phase mixing. opx = orthopyroxene, cpx = clinopyroxene, pl = plagioclase, amph = amphibole, qtz = quartz, gt = garnet, zo = zoisite.







**Figure 6:** SEM-BSE images representative of microstructures documented in deformed Fe-rich opx + plag assemblages. Amph and plag$_2$ are the main reaction products. At 850 °C, the deformation is heterogeneously distributed, and a strain gradient is clearly apparent at sample scale. (a-b) In strongly deformed parts, the amph appears as partially connected shear bands, while in the other parts (c) it appears as thick

coronas around the opx1 and plag1 clasts. (e-f) At 900 °C, the deformation is more homogeneously distributed, and the reaction products appear as reaction coronas. opx = orthopyroxene, cpx = clinopyroxene, pl = plagioclase, amph = amphibole.





**Figure 7:** SEM-BSE images representative of microstructures shown in deformed amph + plag and pure amph assemblages. (a-b-c) Nucleation of amph-rich layers in the amph + plag assemblage deformed at 800 °C. (d-e) The pure amph assemblages deformed at 800 °C and 900 °C show the development of fine-grained mixture zones of amph$_2$ and cpx. opx = orthopyroxene, cpx = clinopyroxene, pl = plagioclase, amph = amphibole, gt = garnet, zo = zoisite.












**Figure 8:** Mineral composition plots. Plagioclase compositions in the Fe-rich opx + plag (a) and Mg-rich opx + plag (b) assemblages plotted on the ternary diagram of orthoclase ($KAlSi_3O_8$), albite ($NaAlSi_3O_8$) and anorthite ($CaAlSi_3O_8$). (c) Pyroxene compositions plotted on the ternary diagram of wollastonite ($Ca_2Si_2O_6$), enstatite ($Mg_2Si_2O_6$) and ferrosilite ($Fe_2Si_2O_6$). Chemical composition of plagioclase and pyroxene are divided into three subsets: clast-core, clast-rim and fine grains. (d) Classification of amphiboles in a $Mg/(Mg + Fe^{2+})$ versus Si content graph, for the case of $Ca \geq 1.5$; $(Na + K)_A <0.5$; $Ca_A <0.5$.



**Figure 9:** Representative Fourier transform infrared (FTIR) spectra of orthopyroxene and plagioclase starting materials. m = average water content.







**Figure 10:** SEM-BSE images of incipient nucleation and shear localization. (a) In the Mg-rich opx + plag assemblages, fine-grained tails of mixed phases nucleate at the edges of original grains. In the Fe-rich opx + plag (b) and amph + plag (c) assemblages, the nucleation is fairly monophase. The new grains tend to organize into anastomosing network of thin amphibole and zoisite (orange triangle). opx = orthopyroxene, pl = plagioclase, amph = amphibole, zo = zoisite.

**Figure 11:** Grain size distribution reported as histogram of grain size versus density for different category of grains in the Mg-opx + plag (a), pure amph (b), Fe-opx + plag (c), and amph + plag (d) assemblages. Overall, there is a significant and systematic reduction in grain size between the original grains and the reaction products. The log normal distribution curves are fit for each grain size distribution. The modal proportion (vol.%) of reaction products and original grains of each assemblage are also reported.





**Figure 12:** SPO and Amph LPOs. (a-b) Rose diagrams of the SPO of plag$_1$, opx$_1$ and amph in the Fe-opx + plag assemblages (a: 533NM— 850 °C and b: 532NM—900 °C). (c) Strongly deformed part of the Fe-opx + plag assemblage deformed at 850 °C. (d) Amph LPOs are shown for the (100) plane, (010) plane and [001] axis with respect to the shear direction, considering one point per grain. pfJ = pole figure texture index, MUD = multiple of uniform distribution, opx = orthopyroxene, pl = plagioclase, amph = amphibole.



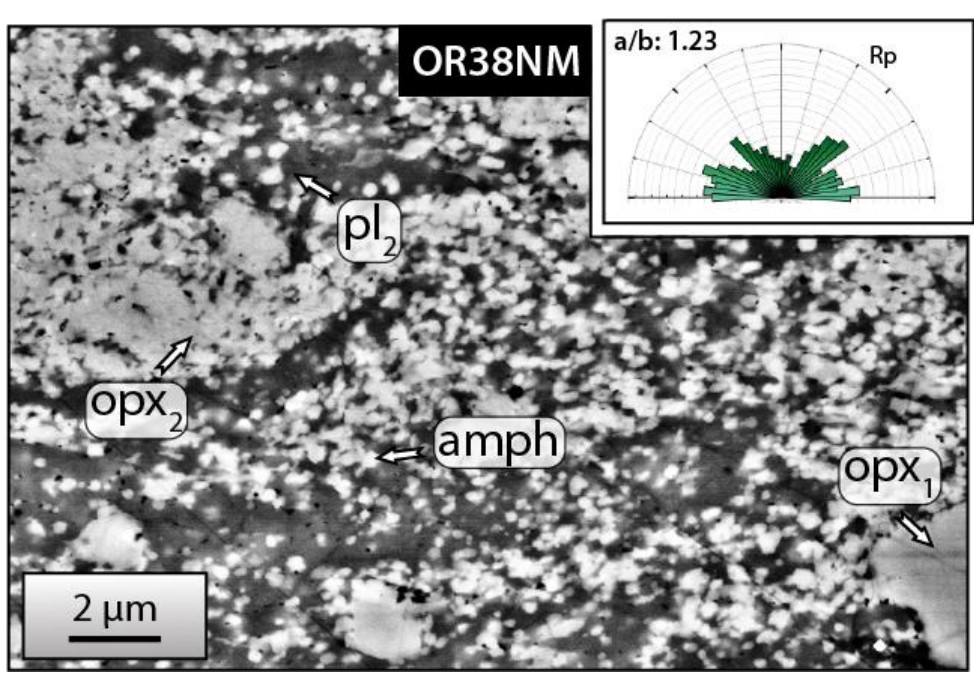

**Figure 13:** Rose diagram of the SPO of undifferentiated reaction products in fine-grained mixed zones in the Mg-rich opx + plag assemblages. opx = orthopyroxene, pl = plagioclase, amph = amphibole.



**Figure 14:** Schematic mechanical and microstructural evolution of the deformed Mg-rich opx + plag and Fe-rich opx + plag assemblages. Rp = reaction product, opx = orthopyroxene, pl = plagioclase, amph = amphibole.





|  | Sonora labradorite | | Damaping Mg-Opx | |
| --- | --- | --- | --- | --- |
|  | Wt. % oxides | Ions per 8 O | Wt. % oxides | Ions per 6 O |
| $SiO_2$ | 53,87 | 2,434 | 55,71 | 1,929 |
| $Al_2O_3$ | 29,41 | 1,566 | 3,81 | 0,155 |
| CaO | 11,68 | 0,565 | 0,42 | 0,016 |
| $Na_2O$ | 4,06 | 0,356 | 0,07 | 0,005 |
| $K_2O$ | 0,46 | 0,027 | 0,02 | 0,001 |
| MgO | 0,09 | 0,006 | 32,51 | 1,678 |
| $TiO_2$ | 0,08 | 0,003 | 0,07 | 0,002 |
| FeO | 0,38 | 0,014 | 7,21 | 0,209 |
| MnO | 0,05 | 0,002 | 0,18 | 0,005 |
| **Total** | 100,08 | 4,972 | 100,01 | 4,000 |

|  | | | |
| --- | --- | --- | --- |
| **An**60 | | **Wo**1 | |
| **Ab**38 | | **En**88 | |
| **Or**2 | | **Fs**11 | |

|  | Hidaka Plag | | Hidaka Fe-Opx | |
| --- | --- | --- | --- | --- |
|  | Wt. % oxides | Ions per 8 O | Wt. % oxides | Ions per 6 O |
| $SiO_2$ | 54,71 | 2,448 | 52,86 | 1,952 |
| $Al_2O_3$ | 29,28 | 1,544 | 1,00 | 0,044 |
| CaO | 11,42 | 0,548 | 1,23 | 0,049 |
| $Na_2O$ | 4,94 | 0,429 | 0,06 | 0,004 |
| $K_2O$ | 0,22 | 0,013 | 0,02 | 0,001 |
| MgO | 0,01 | 0,001 | 22,41 | 1,234 |
| $TiO_2$ | 0,00 | 0,000 | 0,21 | 0,006 |
| FeO | 0,39 | 0,015 | 22,55 | 0,696 |
| MnO | 0,09 | 0,003 | 0,46 | 0,014 |
| **Total** | 100,06 | 5,000 | 100,80 | 4,000 |

|  | | | |
| --- | --- | --- | --- |
| **An**55 | | **Wo**2 | |
| **Ab**44 | | **En**62 | |
| **Or**1 | | **Fs**36 | |

|  | Massif Central Amph | |
| --- | --- | --- |
|  | Wt. % oxides | Ions per 23 O |
| $SiO_2$ | 43,99 | 6,646 |
| $Al_2O_3$ | 9,91 | 1,765 |
| CaO | 11,12 | 1,800 |
| $Na_2O$ | 1,67 | 0,489 |
| $K_2O$ | 0,47 | 0,091 |
| MgO | 10,87 | 2,448 |
| $TiO_2$ | 1,21 | 0,138 |
| FeO | 17,79 | 2,248 |
| MnO | 0,38 | 0,000 |
| **Total** | 97,41 | 15,625 |

**Magnesiohornblende**

**Table 1:** Chemical compositions of plagioclase, pyroxene and amphibole starting materials.





1315

| Exp. Nr | Material | Type | T | P | H₂O | $\tau_{peak}$ | $\tau_{flow}$ | $\tau_{end}$ | γ | th₀ | th_f | t |
|---------|----------|------|-----|------|------|--------|--------|--------|-----|------|------|-----|
| | | | [°C] | [GPa] | µL | [MPa] | [MPa] | [MPa] | | [mm] | [mm] | [h] |
| 557NM | Mg-Opx + Plag | PS[x] | 850 | 1 | 0,12 | 1067 | - | 577 | 0,4 | 0,75 | 0,59 | 35 |
| 559NM | Mg-Opx + Plag | PS[x] | 800 | 1 | 0,12 | 1111 | - | 350 | 0,5 | 0,75 | 0,67 | 38 |
| OR24NM | Mg-Opx + Plag | D[x] | 800 | 1 | 0,25 | 1280 | - | 866 | 3,1 | 1,1 | 0,87 | 85 |
| OR34NM | Mg-Opx + Plag | D | 900 | 1 | 0,25 | 781 | 114 | 126 | 7,6 | 1,1 | 0,68 | 96 |
| OR38NM | Mg-Opx + Plag | D | 850 | 1 | 0,25 | 1037 | 339 | 339 | 8,0 | 1,1 | 0,63 | 83 |
| OR41NM | Mg-Opx + Plag | D | 900 | 1 | 0,25 | 1094 | 542 | 542 | 7,0 | 1,1 | 0,72 | 53 |
| OR47NM | Mg-Opx + Plag | PS | 900 | 1 | 0,25 | 989 | - | 989 | 0,6 | 1,1 | 0,91 | 25 |
| OR49NM | Mg-Opx + Plag | D[x] | 900 | 1 | 0,25 | 1111 | 800 | 800 | 6,0 | 1,1 | 0,73 | 68 |
| OR51NM | Mg-Opx + Plag | PS | 900 | 1 | 0,25 | 901 | - | 901 | 0,8 | 1,1 | 1 | 156 |
| 532NM | Fe-Opx + Plag | D | 900 | 1 | 0,12 | - | - | 405 | 5,0 | 0,75 | 0,54 | 38 |
| 533NM | Fe-Opx + Plag | D | 850 | 1 | 0,12 | 645 | 683 | 683 | 3,8 | 0,75 | 0,47 | 50 |
| 538NM | Fe-Opx + Plag | D[x] | 800 | 1 | 0,12 | 1080 | - | 563 | 0,9 | 0,75 | 0,73 | 28 |
| OR61NM | Fe-Opx + Plag | PS | 850 | 1 | 0,25 | 885 | - | 897 | 1,7 | 1,1 | 0,73 | 77 |
| OR15NM | Amph | D | 900 | 1 | 0,20 | 348 | 147 | 147 | 3,7 | 0,9 | 0,62 | 104 |
| OR18NM | Amph | D | 800 | 1 | 0,20 | 803 | 370 | 370 | 5,4 | 0,9 | 0,68 | 73 |
| OR23NM | Amph + Plag | D | 800 | 1 | 0,20 | 563 | 435 | 483 | 5,8 | 0,9 | 0,60 | 31 |

**Table 2:** List of experiments and experimental conditions. Type: PS: peak stress, D: deformed samples to varying amounts of shear strain. A cross is added to the samples where the forcing blocks started to slip at the sample interface; $\tau_{peak}$: differential stress at peak, $\tau_{flow}$: steady-state differential stress, $\tau_{end}$: differential stress at end of experiment, γ: shear strain, th₀: thickness initial of the shear zone, th_f: final shear zone thickness, t: time before hit point.