# Peer review of "Relationship between microstructures and resistance in mafic assemblages that deform and transform."

_Solid Earth, 2020_

## Referee Comment (RC1) · Jolien Linckens (Referee) · 8 Jul 2020

In this manuscript the authors describe shear deformation experiments performed on mafic rocks with four different compositions. The authors infer that if reaction and nucleation rates in the assemblages are fast, the experiments show no peak stress and deform at lower stresses than experiments where the nucleation rates are slower. A faster nucleation of reaction products is assumed to be caused by the presence of water inside the minerals of the starting material. When the reaction products form fine-grained interconnected layers the experiments show a strain weakening. In contrast, when the reaction products are coarser grained and have a poor connectivity

the experiments show a steady-state or a strain hardening. The experiments highlight the importance of reactions during deformation. The results indicate that the ability of minerals to react will determine where the strain is localized. With ongoing strain, the grain size and connectivity of the reaction products will determine if a large weakening occurs.

The manuscript is well-written and of excellent scientific quality. The results, discussion and conclusion are presented in a clear, concise and structured way. The quality of the figures are high. I have some minor comments below, and I suggest the manuscript is accepted with minor revisions. I hope the comments are useful.

Line 175: Why did you do experiments on the Mg-rich assemblages with a long and a short run-in time? In the results you mention that the experiments with a longer run-in show systematic weakening after peak stress. I guess this is because the assemblages have a longer time to react? This is later not mentioned in the discussion.

Line 489-490: did you compare the experiments of Fe-rich and Mg-rich assemblages at low shear strain to see if indeed the Fe-rich samples contain more reaction products at the beginning of the experiment? This would strengthen your argument that the lack of peak stress and initial lower stresses in the Fe-rich samples is due to the faster nucleation rate in these samples. Related to this comment: can the initial lower stresses in the Fe-rich assemblages not be due just to the higher water content inside the starting minerals compared to the dry minerals in the Mg-rich assemblages?

Minor comments: Line 216-217: you talk here about the amph rich shear bands in the Fe-rich opx + plag assemblage but then refer to figure 3c which is belonging to the amph + plag assemblage. Line 478: how did you estimate the amphibole content in the samples? This is not mentioned in the methods. Line 481: for completion it would be nice to have a number of the amphibole content in the Mg-opx + plag assemblage. In Fig 5b, d and f it seems to be quite an amount. Fig. 11: How did you measure the opx2+plag2+amph grain sizes in the Mg-rich opx + plag samples? Did you use

[Figure]

EBSD maps as well for this, in the methods you mentioned you only use EBSD maps to determine the amph grain size. When I look at fig 10a it is not clear to me how you can determine the grain size from these BSE images. The same for the cpx in the amph + plag and pure amph samples.
* * *

---

## Referee Comment (RC2) · Luca Menegon (Referee) · 5 Aug 2020

Review of "Relationship between microstructures and resistance in mafic assemblages that deform and transform", by Mansard et al.

Dear Authors and Editor, First, I apologize for the late submission of my review report.

This manuscript presents a detailed experimental study of the feedback between mineral reactions and deformation in "wet" mafic assemblages deformed under high P, T conditions with a Griggs-type solid medium apparatus. The experimental samples have been investigated in detail with electron microscopy (including EBSD) and image

analysis techniques.

The work aims to test the role of synkinematic mineral reactions on the rheology of mafic assemblages of different compositions. Depending on different rates of reaction progress and of the associated microstructural development, the stress-strain behaviour and the extent of weakening varies in the different assemblages. The results highlight that differences in mechanical strength depend on the microstructural evolution of the assemblage, which in turn is determined by the rate and the type of synkinematic mineral reactions. The Authors assume that faster reaction rates depend on the higher intracrystalline water content in the starting material. The results are also discussed in terms of the strain localization potential of pyroxene vs amphibole-dominated mafic assemblages.

The conclusions are largely supported by the results, and further highlight the fundamental feedback between mineral reactions, deformation, and strain localization. The paper is very well written and illustrated, the experimental work and the microstructural analysis are meticulous, and the overall dataset is of high quality. I definitely recommend this article for publication in Solid Earth. I have only a few suggestions for minor revisions, keyed to line numbers. Congratulations to the Authors on this very good piece of work.

Line 62: if I may, I suggest to add the work by degli Alessandrini et al (Lithos 2017), as it investigated in detail the effect of reactions on the rheology of pyroxene-bearing mafic assemblages deformed at lower crustal conditions.

Line 136: please add information on the grain size of the starting material to justify the spot size of 40 x 40 mm2 used for the FTIR analysis. Many grains of the starting material look considerably smaller than this sport size in the BSE images.

Line 221: it might be correct that mineral reactions preferentially occur in strongly deformed areas, but likewise it might be that layers of reaction products (that originally nucleated in a different position) are transposed and smeared off along the foliation.

This is typically the case, for instance, in recrystallized myrmekite (see Ceccato et al., 2018). I would argue that mineral reactions in shear zones tend to form at sites of stress (and elastic strain) concentrations (which are typically those facing the instantaneous shortening axis), so that perhaps low-strain samples are more appropriate to identify the nucleation sites of mineral reactions.

Lines 226-232: the cpx-forming reactions in the amph-plag assemblages are dehydration reactions, which typically result in the formation of melt even at 800-900 °C (e.g., Wolff and Willie, 1994). Is there any microstructural evidence of melt pseudomorphs, and has the melt-in curve been calculated in the thermodynamic modelling in order to ensure that the experiments were performed fully into the solidus field?

Lines 276-282: please add sketches of the SC-SC' fabrics in Figs. 10a-c to better summarize these observations.

Line 303: whilst the SC' fabric is clear in Fig. 12c, Fig. 12a looks more an SC fabric. Please add sketches/annotations to highlight the fabric elements.

Line 312: I understand that the pole figures are plotted as one point per grain; please provide the total number of the plotted grains, the step size and the average grain size of amphibole, so that the reader can make their judgement on the data acquisition and processing routines. How many data points did you consider representative to define an individual "grain"? Amphibole grain size in Fig. 13 looks < 1 micron, so I wonder whether many of you "grains" are actually individual data points that might encompass more than one single grain. Please clarify.

Line 360: is there any evidence of dislocation creep been potentially active in the strong phases? Do you have EBSD maps of porphyroclasts that could help understand this?

Line 400: please see my comment to line 221. Perhaps the reaction products nucleated elsewhere and were transposed/smeared along the porphyroclast tails with increasing strain. From some BSE images, it seems that the entire porphyroclasts are locally

rimmed by reaction products (e.g., Fig. 7c, Figs. 10), so I wonder what the original nucleation site was.

Lines 494-497: this is a very interesting and plausible interpretation. But the follow up question is how did the H2O stored in the interior of strong porphyroclasts become available for the reactions? Did microfracturing play a role here? Any evidence?

Line 532: syn-kinematic mineral reactions are very important for the deformation of mafic systems also at higher metamorphic grades (see degli Alessandrini et al., 2017). Here you also document dehydration reactions and their role on deformation.

With best wishes, Luca Menegon

---

## Author Comment (AC1) · 25 Aug 2020

Author's response to comments from Reviewer 1 (J.L.)

General comments Rev.1: In this manuscript the authors describe shear deformation experiments performed on mafic rocks with four different compositions. The authors infer that if reaction and nucleation rates in the assemblages are fast, the experiments show no peak stress and deform at lower stresses than experiments where the nucleation rates are slower. A faster nucleation of reaction products is assumed to be caused by the presence of water inside the minerals of the starting material. When the reaction products form fine-grained interconnected layers the experiments show

a strain weakening. In contrast, when the reaction products are coarser grained and have a poor connectivity the experiments show a steady-state or a strain hardening. The experiments highlight the importance of reactions during deformation. The results indicate that the ability of minerals to react will determine where the strain is localized. With ongoing strain, the grain size and connectivity of the reaction products will determine if a large weakening occurs. The manuscript is well-written and of excellent scientific quality. The results, discussion and conclusion are presented in a clear, concise and structured way. The quality of the figures are high. I have some minor comments below, and I suggest the manuscript is accepted with minor revisions. I hope the comments are useful.

Authors: We would like to thank you for the suggestions to improve the manuscript and greatly appreciate the comments and suggestions of the manuscript.

—

Rev.1: Line 175: Why did you do experiments on the Mg-rich assemblages with a long and a short run-in time? In the results you mention that the experiments with a longer run-in show systematic weakening after peak stress. I guess this is because the assemblages have a longer time to react? This is later not mentioned in the discussion.

Authors: These experiments have been carried out in order to be able to compare experiments reaching similar shear strain with different total durations. We document that the longer run-in periods cause reaction products to nucleate before deformation starts, whereas shorter run-in periods cause reaction products to form only during deformation. Thus, carrying out these experiments allows us to study the relationship between the amount of reaction products over time and the effect of deformation. This has been documented in our recently accepted article (Mansard et al., 2020), and we have added a sentence to explain the situation (lines 183-184).

—

Rev.1: Line 489-490: did you compare the experiments of Fe-rich and Mg-rich assemblages at low shear strain to see if indeed the Fe-rich samples contain more reaction products at the beginning of the experiment? This would strengthen your argument that the lack of peak stress and initial lower stresses in the Fe-rich samples is due to the faster nucleation rate in these samples. Related to this comment: can the initial lower stresses in the Fe-rich assemblages not be due just to the higher water content inside the starting minerals compared to the dry minerals in the Mg-rich assemblages?

Authors: The reviewer is right, a comparison of samples at low strain would contribute to clarify whether reaction progress at the early stages affects the strength. We do not have samples with the same low shear strain, but some approximate comparison is possible. We have added a sentence explaining the difference in the text (lines 501-503). Considering that the rheological behavior is strongly controlled by the reaction products, the Fe-rich opx + plag assemblages initially deform at lower stresses than the Mg-rich opx + plag assemblages because in the Fe-rich assemblages reaction products nucleate faster than in the Mg-rich assemblages at early stages of the experiment. The explanation of a lower strength due to different $H_2O$ content does not work because deformation is not accommodated by crystal plasticity in these samples. We have added a sentence explaining this fact, too (lines 513-514).

—

Rev.1: Line 216-217: you talk here about the amph rich shear bands in the Fe-rich opx + plag assemblage but then refer to figure 3c which is belonging to the amph + plag assemblage.

Authors: The reviewer is right. We have changed the sentence to refer to the right figure (lines 226-227).

—

Rev.1: Line 478: how did you estimate the amphibole content in the samples? This is

not mentioned in the methods.

Authors: We have added a description of the method to the text (lines 161-168): In order to estimate the proportion of phases we used SEM/BSE images to produce manually digitized grain maps with the illustrator software, when it was possible to distinguish the grain boundaries (e.g. Fe-rich opx + plag assemblages). From these phase maps we could separate the phases with the ImageJ software and estimate their proportion. This is how the amount of amphibole in the Fe-rich opx + plag assemblages is estimated. When it was impossible to distinguish the grains individually, we drew areas that corresponded either to a set of grains of the same phase or to several phases that could not be separated. Some grains are too small to be separated with enough confidence from other grains. For this reason, we have included all reaction products together, and have not separated the amphibole from the plagioclase2 and the pyroxene2 in the Mg-rich opx + plag assemblages. Section 2.3 has been modified to clarify that the proportion of phases is estimated from the manually digitized grain maps by using the software ImageJ.

—

Rev.1: Line 481: for completion it would be nice to have a number of the amphibole content in the Mg-opx + plag assemblage. In Fig 5b, d and f it seems to be quite an amount. Fig. 11: How did you measure the opx2+plag2+amph grain sizes in the Mg-rich opx + plag samples? Did you use EBSD maps as well for this, in the methods you mentioned you only use EBSD maps to determine the amph grain size. When I look at fig 10a it is not clear to me how you can determine the grain size from these BSE images. The same for the cpx in the amph + plag and pure amph samples.

Authors: The reviewer is right, it would be nice to have number of the amphibole content in the Mg-rich opx + plag assemblages. However, we cannot be more precise because these phases are difficult to threshold on a large scale (cf. previous comment). Although locally we are able to distinguish individual grains, this is still on a small scale

(Fig. 5b-d). This is why it may be noticed that throughout this manuscript, especially in Figure 11, we group the amphibole with pyroxene (opx2) when we are attempting to estimate proportions of reaction products in the Mg-rich opx + plag assemblages. Regarding the amphibole grain size, we can easily estimate it with our manually digitized grain maps in the Fe-rich opx + plag assemblages. On the other hand, as mentioned, we need to use EBSD maps to determine the amphibole grain size when it is not too small. To give an example, the grain sizes in Figure 13 are too small to be analyzed at EBSD. Only rigorous observation at the SEM and rigorous segmentation of the grains can then allow estimation of the grain size (e.g. Heilbronner and Barrett, 2005 – Image Analysis in Earth Sciences).

[Figure]

---

## Author Comment (AC2) · 25 Aug 2020

Author's response to comments from Reviewer 2 (L.M.)

General comments

Rev.2: This manuscript presents a detailed experimental study of the feedback between mineral reactions and deformation in "wet" mafic assemblages deformed under high P, T conditions with a Griggs-type solid medium apparatus. The experimental samples have been investigated in detail with electron microscopy (including EBSD) and image analysis techniques. The work aims to test the role of synkinematic mineral

reactions on the rheology of mafic assemblages of different compositions. Depending on different rates of reaction progress and of the associated microstructural development, the stress-strain behaviour and the extent of weakening varies in the different assemblages. The results highlight that differences in mechanical strength depend on the microstructural evolution of the assemblage, which in turn is determined by the rate and the type of synkinematic mineral reactions. The Authors assume that faster reaction rates depend on the higher intracrystalline water content in the starting material. The results are also discussed in terms of the strain localization potential of pyroxene vs amphibole-dominated mafic assemblages. The conclusions are largely supported by the results, and further highlight the fundamental feedback between mineral reactions, deformation, and strain localization. The paper is very well written and illustrated, the experimental work and the microstructural analysis are meticulous, and the overall dataset is of high quality. I definitely recommend this article for publication in Solid Earth. I have only a few suggestions for minor revisions, keyed to line numbers. Congratulations to the Authors on this very good piece of work.

Authors: We are very thankful for the thorough and constructive review. We greatly appreciate the comments and suggestions to the manuscript.

—

Rev.2: Line 62: if I may, I suggest to add the work by degli Alessandrini et al (Lithos 2017), as it investigated in detail the effect of reactions on the rheology of pyroxene-bearing mafic assemblages deformed at lower crustal conditions.

Authors: Correct, this is a good suggestion. We have cited the paper and have added this work to the list of references (line 62-63).

—

Rev.2: Line 136: please add information on the grain size of the starting material to justify the spot size of 40 x 40 mm2 used for the FTIR analysis. Many grains of the

starting material look considerably smaller than this sport size in the BSE images.

Authors: We thank the reviewer for pointing this out, it was not very clear in the text. Double-polished thick sections ($\sim$150–200 $\mu$m) of the starting materials were prepared for FTIR analysis. These thick sections were prepared from mineral powders for the Mg-rich opx + plag sample and from a natural section of mylonite for the Fe-rich opx + plag sample. In both cases, the grain size used for FTIR analysis is much larger (>100 $\mu$m) than the one we decided to use in our experiments (between 10 and 20 $\mu$m). Many grains would indeed be much too small for the spot size of 40 $\times$ 40 $\mu$m2. The text has been modified accordingly (lines 133-139).

—

Rev.2: Line 221: it might be correct that mineral reactions preferentially occur in strongly deformed areas, but likewise it might be that layers of reaction products (that originally nucleated in a different position) are transposed and smeared off along the foliation. This is typically the case, for instance, in recrystallized myrmekite (see Ceccato et al., 2018). I would argue that mineral reactions in shear zones tend to form at sites of stress (and elastic strain) concentrations (which are typically those facing the instantaneous shortening axis), so that perhaps low-strain samples are more appropriate to identify the nucleation sites of mineral reactions.

Rev.2: Line 400: please see my comment to line 221. Perhaps the reaction products nucleated elsewhere and were transposed/smeared along the porphyroclast tails with increasing strain. From some BSE images, it seems that the entire porphyroclasts are locally rimmed by reaction products (e.g., Fig. 7c, Figs. 10), so I wonder what the original nucleation site was.

Authors: The two previous comments are related, so we decided to group them together to avoid repetition. We have recently published a paper that discusses in more detail these aspects of original nucleation in the early stages of deformation in the Mg-rich opx + plag assemblages (Mansard et al., 2020 – JSG). In hot-pressing exper-

iments, we document that the reaction products occur as thin coronas at the Opx1-Plag1 phase boundaries or in cracks. The rims grow concentrically around old grain relicts without any specific locations at Opx-Plag phase boundaries. At peak stress, little change compared to the hot-pressing experiments, although the amount of reaction products increases slightly and start to coalesce to form partially connected aggregates. At intermediate shear strain, we observed the development of subparallel fine-grained polyphase shear bands, originating from tails that extend from the edges of original Opx, and progressively coalesce to form an interconnected network. Thus, the phase mixing starts at the edges of the original Opx that is gradually consumed by the reaction as evidenced by irregular Opx boundaries, where new grains nucleate along low-stress sites. With increasing strain, these "thin" shear bands evolve into broader high-strain zones. As the reviewer's comment is a discussion point and the microstructures shown in this manuscript do not provide any evidence for the reviewer's point of view, we have not modified the text.

—

Rev.2: Line 226-232: the cpx-forming reactions in the amph-plag assemblages are dehydration reactions, which typically result in the formation of melt even at 800-900°C (e.g., Wolff and Willie, 1994). Is there any microstructural evidence of melt pseudomorphs, and has the melt-in curve been calculated in the thermodynamic modelling in order to ensure that the experiments were performed fully into the solidus field?

Authors: This is a very important point, given that melt reactions may strongly influence the mechanical behaviour of rocks. Indeed, we had considered the possibility of the formation of melt given these experimental conditions of pressure and temperature, However, after extensive observation at the SEM, we did not find any microstructural evidence of melt pockets or pseudomorphs. We have added a sentence about that to the text (lines 242-243). In addition, according to thermodynamic modeling, the P-T conditions should be outside the melt-forming field.

[Figure]

—

Rev.2: Line 276-282: please add sketches of the SC-SC' fabrics in Figs. 10a-c to better summarize these observations.

Rev.2: Line 303: whilst the SC' fabric is clear in Fig. 12c, Fig. 12a looks more an SC fabric. Please add sketches/annotations to highlight the fabric elements.

Authors: Here again, we decided to group the two previous comments together to avoid repetition. Following your advice, we have added annotations to several figures to highlight the fabric elements (Figs. 4a ; 6a-b ; 7a ; 12a-b-c). In the Mg-rich opx + plag samples, strain tends to produce mixed and connected fine-grained bands with C-type geometry in high-strain zones (e.g. Fig. 10a). In contrast, the Fe-rich opx + plag and amph + plag mixtures tend to form clusters and only locally connected amph-rich rich $\sigma$-tails at porphyroclasts, as S-C- (e.g. Fig 12a-b) or S-C'- (Fig. 12c) type geometries.

—

Rev.2: Line 312: I understand that the pole figures are plotted as one point per grain; please provide the total number of the plotted grains, the step size and the average grain size of amphibole, so that the reader can make their judgement on the data acquisition and processing routines. How many data points did you consider representative to define an individual "grain"? Amphibole grain size in Fig. 13 looks < 1 micron, so I wonder whether many of you "grains" are actually individual data points that might encompass more than one single grain. Please clarify.

Authors: The reviewer has pointed out an important issue, we have added the data you mentioned (see caption of Figure 12 – line 1272). Then, we consider that 5 data points are required and are considered representative to define an individual grain. We have added a sentence about that to the methods section (line 151). With regard to the grain size in Figure 13, it is indeed extremely small. However, this figure is for the Mg-rich opx + plag assemblages. The pole figures in Figure 12 are associated only with the

Fe-rich opx + plag assemblages. Unfortunately, none of our EBSD analyses performed on Mg-rich opx + plag assemblages were convincing, due in part to the extremely small grain sizes in the mixture zones. To clarify, we have indicated in the caption that the data presented in Figure 12 are only associated to the Fe-rich opx + plag assemblages (line 1269).

—

Rev.2: Line 360: is there any evidence of dislocation creep been potentially active in the strong phases? Do you have EBSD maps of porphyroclasts that could help understand this?

Authors: Unfortunately, none of our EBSD sessions on Mg-rich opx + plag assemblages were conclusive. This is most certainly due to the fact that we tried to study very fine grains within a narrow shear zone trapped between 2 very strong alumina pistons, thus making polishing very complex. The grain shapes of porphyropclasts do not suggest any deformation by crystal plasticity.

—

Rev.2: Line 494-497: this is a very interesting and plausible interpretation. But the follow up question is how did the H2O stored in the interior of strong porphyroclasts become available for the reactions? Did microfracturing play a role here? Any evidence?

Authors: The reviewer has made an impotant point here. Indeed, cracking in opx is very common : original Opx clasts are locally cut by brittle fractures which refines the grain size of strong porphyroclasts. The cracking will make the H2O available for reactions. We have added a sentence to the text to say this (lines 509-511).

—

Rev.2: Line 532: syn-kinematic mineral reactions are very important for the deformation of mafic systems also at higher metamorphic grades (see degli Alessandrini et al.,

2017). Here you also document dehydration reactions and their role on deformation.

Authors: The reviewer is right. We have added this work to the reference list (line 550).

---

## Author Response (AR2)

**RESPONSE TO EDITOR AND REVIEWER**

Dear Editor and Reviewer,

5   The authors would like to thank the editor and reviewer for the opportunity to revise our paper on "Relationship between microstructures and resistance in mafic assemblages that deform and transform". We wish to thank the editor and reviewer for their precious time and invaluable comments. The suggestions offered have been very helpful for improving our manuscript.

We have prepared a new version of the manuscript including most of the reviewer's suggestions. We have enriched the text
10   with new data, information and references.

We have included the reviewer comments below this letter and responded to them individually and describing changes we have made in the manuscript. Revisions have been approved by the four authors.

15   Yours sincerely,
On behalf of the co-authors,
Nicolas MANSARD

**Authors' response to the editor's comment.**

**Comments to the Author:**

Dear Dr Mansard and coauthors,

Thank you for your careful revision of the manuscript and your detailed engagement with the reviewers' comments. The manuscript reads really well now.

I did note that a reader who is less versed in the metamorphic reactions involved might benefit from a brief paragraph
40 introducing the principal reactions you expect in the system. I am not asking for an actual analysis, I am aware that the very small grain size precludes an exact compositional analysis of the actual reaction products, and this also does not seem critical for the principal findings of the manuscript. It would be good though to write briefly what you expect to happen, to lay the path for the findings you report/discuss later on.

45 With best regards,
Florian Fusseis

**Authors**: Thank you very much for your comment and your positive evaluation of the manuscript. Following your advice, we have added a paragraph at the beginning of the section "mineral reactions and microstructures" (lines 474-480).

We would like to thank you again for your editorial handling.

With best regards,
Nicolas Mansard
* * *
**Author's response to comments from Reviewer 1 (J.L.)**

**General comments**

60 *Rev.1*: In this manuscript the authors describe shear deformation experiments performed on mafic rocks with four different compositions. The authors infer that if reaction and nucleation rates in the assemblages are fast, the experiments show no peak stress and deform at lower stresses than experiments where the nucleation rates are slower. A faster nucleation of reaction products is assumed to be caused by the presence of water inside the minerals of the starting material. When the reaction products form fine-grained interconnected layers the experiments show a strain weakening. In contrast, when the reaction
65 products are coarser grained and have a poor connectivity the experiments show a steady-state or a strain hardening. The

experiments highlight the importance of reactions during deformation. The results indicate that the ability of minerals to react will determine where the strain is localized. With ongoing strain, the grain size and connectivity of the reaction products will determine if a large weakening occurs. The manuscript is well-written and of excellent scientific quality. The results, discussion and conclusion are presented in a clear, concise and structured way. The quality of the figures are high. I have some minor comments below, and I suggest the manuscript is accepted with minor revisions. I hope the comments are useful.

**Authors:** We would like to thank you for the suggestions to improve the manuscript and greatly appreciate the comments and suggestions of the manuscript.
* * *
**Rev.1:** Line 175: Why did you do experiments on the Mg-rich assemblages with a long and a short run-in time? In the results you mention that the experiments with a longer run-in show systematic weakening after peak stress. I guess this is because the assemblages have a longer time to react? This is later not mentioned in the discussion.

**Authors:** These experiments have been carried out in order to be able to compare experiments reaching similar shear strain with different total durations. We document that the longer run-in periods cause reaction products to nucleate before deformation starts, whereas shorter run-in periods cause reaction products to form only during deformation. Thus, carrying out these experiments allows us to study the relationship between the amount of reaction products over time and the effect of deformation. This has been documented in our recently accepted article (Mansard et al., 2020), and we have added a sentence to explain the situation (lines 183-184).
* * *
**Rev.1:** Line 489-490: did you compare the experiments of Fe-rich and Mg-rich assemblages at low shear strain to see if indeed the Fe-rich samples contain more reaction products at the beginning of the experiment? This would strengthen your argument that the lack of peak stress and initial lower stresses in the Fe-rich samples is due to the faster nucleation rate in these samples. Related to this comment: can the initial lower stresses in the Fe-rich assemblages not be due just to the higher water content inside the starting minerals compared to the dry minerals in the Mg-rich assemblages?

**Authors:** The reviewer is right, a comparison of samples at low strain would contribute to clarify whether reaction progress at the early stages affects the strength. We do not have samples with the same low shear strain, but some approximate comparison is possible. We have added a sentence explaining the difference in the text (lines 501-503). Considering that the rheological behavior is strongly controlled by the reaction products, the Fe-rich opx + plag assemblages initially deform at lower stresses than the Mg-rich opx + plag assemblages because in the Fe-rich assemblages reaction products nucleate faster than in the Mg-rich assemblages at early stages of the experiment. The explanation of a lower strength due to different $H_2O$ content does not work because deformation is not accommodated by crystal plasticity in these samples. We have added a sentence explaining this fact, too (lines 513-514).

**Rev.1:** Line 216-217: you talk here about the amph rich shear bands in the Fe-rich opx + plag assemblage but then refer to figure 3c which is belonging to the amph + plag assemblage.

**Authors:** The reviewer is right. We have changed the sentence to refer to the right figure (lines 226-227).

105

**Rev.1:** Line 478: how did you estimate the amphibole content in the samples? This is not mentioned in the methods.

**Authors:** We have added a description of the method to the text (lines 161-168):

In order to estimate the proportion of phases we used SEM/BSE images to produce manually digitized grain maps with the illustrator software, when it was possible to distinguish the grain boundaries (e.g. Fe-rich opx + plag assemblages). From these phase maps we could separate the phases with the ImageJ software and estimate their proportion. This is how the amount of amphibole in the Fe-rich opx + plag assemblages is estimated. When it was impossible to distinguish the grains individually, we drew areas that corresponded either to a set of grains of the same phase or to several phases that could not be separated. Some grains are too small to be separated with enough confidence from other grains. For this reason, we have included all reaction products together, and have not separated the amphibole from the plagioclase2 and the pyroxene2 in the Mg-rich opx + plag assemblages. Section 2.3 has been modified to clarify that the proportion of phases is estimated from the manually digitized grain maps by using the software ImageJ.

120 **Rev.1:** Line 481: for completion it would be nice to have a number of the amphibole content in the Mg-opx + plag assemblage. In Fig 5b, d and f it seems to be quite an amount. Fig. 11: How did you measure the opx2+plag2+amph grain sizes in the Mg-rich opx + plag samples? Did you use EBSD maps as well for this, in the methods you mentioned you only use EBSD maps to determine the amph grain size. When I look at fig 10a it is not clear to me how you can determine the grain size from these BSE images. The same for the cpx in the amph + plag and pure amph samples.

125 **Authors:** The reviewer is right, it would be nice to have number of the amphibole content in the Mg-rich opx + plag assemblages. However, we cannot be more precise because these phases are difficult to threshold on a large scale (cf. previous comment). Although locally we are able to distinguish individual grains, this is still on a small scale (Fig. 5b-d). This is why it may be noticed that throughout this manuscript, especially in Figure 11, we group the amphibole with pyroxene (opx2) when we are attempting to estimate proportions of reaction products in the Mg-rich opx + plag assemblages. Regarding the amphibole grain size, we can easily estimate it with our manually digitized grain maps in the Fe-rich opx + plag assemblages. On the other hand, as mentioned, we need to use EBSD maps to determine the amphibole grain size when it is not too small. To give an example, the grain sizes in Figure 13 are too small to be analyzed at EBSD. Only rigorous observation at the SEM

and rigorous segmentation of the grains can then allow estimation of the grain size (e.g. Heilbronner and Barrett, 2005 – Image Analysis in Earth Sciences).

135
* * *
140

**Author's response to comments from Reviewer 2 (L.M.)**

**General comments**

*Rev.2*: This manuscript presents a detailed experimental study of the feedback between mineral reactions and deformation in "wet" mafic assemblages deformed under high P, T conditions with a Griggs-type solid medium apparatus. The experimental samples have been investigated in detail with electron microscopy (including EBSD) and image analysis techniques.

The work aims to test the role of synkinematic mineral reactions on the rheology of mafic assemblages of different compositions. Depending on different rates of reaction progress and of the associated microstructural development, the stress-strain behaviour and the extent of weakening varies in the different assemblages. The results highlight that differences in mechanical strength depend on the microstructural evolution of the assemblage, which in turn is determined by the rate and the type of synkinematic mineral reactions. The Authors assume that faster reaction rates depend on the higher intracrystalline water content in the starting material. The results are also discussed in terms of the strain localization potential of pyroxene vs amphibole-dominated mafic assemblages.

The conclusions are largely supported by the results, and further highlight the fundamental feedback between mineral reactions, deformation, and strain localization. The paper is very well written and illustrated, the experimental work and the microstructural analysis are meticulous, and the overall dataset is of high quality. I definitely recommend this article for publication in Solid Earth. I have only a few suggestions for minor revisions, keyed to line numbers. Congratulations to the Authors on this very good piece of work.

**Authors:** We are very thankful for the thorough and constructive review. We greatly appreciate the comments and suggestions to the manuscript.
* * *
*Rev.2*: Line 62: if I may, I suggest to add the work by degli Alessandrini et al (Lithos 2017), as it investigated in detail the effect of reactions on the rheology of pyroxene-bearing mafic assemblages deformed at lower crustal conditions.

165    **Authors:** Correct, this is a good suggestion. We have cited the paper and have added this work to the list of references (line 62-63).

*Rev.2*: Line 136: please add information on the grain size of the starting material to justify the spot size of 40 x 40 mm2 used
170    for the FTIR analysis. Many grains of the starting material look considerably smaller than this sport size in the BSE images.
    **Authors:** We thank the reviewer for pointing this out, it was not very clear in the text. Double-polished thick sections (~150–200 μm) of the starting materials were prepared for FTIR analysis. These thick sections were prepared from mineral powders for the Mg-rich opx + plag sample and from a natural section of mylonite for the Fe-rich opx + plag sample. In both cases, the grain size used for FTIR analysis is much larger (>100 μm) than the one we decided to use in our experiments (between 10
175    and 20 μm). Many grains would indeed be much too small for the spot size of 40 × 40 μm2. The text has been modified accordingly (lines 133-139).

*Rev.2*: Line 221: it might be correct that mineral reactions preferentially occur in strongly deformed areas, but likewise it might
180    be that layers of reaction products (that originally nucleated in a different position) are transposed and smeared off along the foliation. This is typically the case, for instance, in recrystallized myrmekite (see Ceccato et al., 2018). I would argue that mineral reactions in shear zones tend to form at sites of stress (and elastic strain) concentrations (which are typically those facing the instantaneous shortening axis), so that perhaps low-strain samples are more appropriate to identify the nucleation sites of mineral reactions.
185    *Rev.2*: Line 400: please see my comment to line 221. Perhaps the reaction products nucleated elsewhere and were transposed/smeared along the porphyroclast tails with increasing strain. From some BSE images, it seems that the entire porphyroclasts are locally rimmed by reaction products (e.g., Fig. 7c, Figs. 10), so I wonder what the original nucleation site was.

190    **Authors:** The two previous comments are related, so we decided to group them together to avoid repetition. We have recently published a paper that discusses in more detail these aspects of original nucleation in the early stages of deformation in the Mg-rich opx + plag assemblages (Mansard et al., 2020 – JSG). In hot-pressing experiments, we document that the reaction products occur as thin coronas at the $Opx_1$-$Plag_1$ phase boundaries or in cracks. The rims grow concentrically around old grain relicts without any specific locations at Opx-Plag phase boundaries. At peak stress, little change compared to the hot-pressing
195    experiments, although the amount of reaction products increases slightly and start to coalesce to form partially connected aggregates. At intermediate shear strain, we observed the development of subparallel fine-grained polyphase shear bands, originating from tails that extend from the edges of original Opx, and progressively coalesce to form an interconnected network. Thus, the phase mixing starts at the edges of the original Opx that is gradually consumed by the reaction as evidenced

by irregular Opx boundaries, where new grains nucleate along low-stress sites. With increasing strain, these "thin" shear bands

200 evolve into broader high-strain zones. As the reviewer´s comment is a discussion point and the microstructures shown in this manuscript do not provide any evidence for the reviewer´s point of view, we have not modified the text.
* * *
***Rev.2***: Line 226-232: the cpx-forming reactions in the amph-plag assemblages are dehydration reactions, which typically result in the formation of melt even at 800-900°C (e.g., Wolff and Willie, 1994). Is there any microstructural evidence of melt

205 pseudomorphs, and has the melt-in curve been calculated in the thermodynamic modelling in order to ensure that the experiments were performed fully into the solidus field?

**Authors:** This is a very important point, given that melt reactions may strongly influence the mechanical behaviour of rocks. Indeed, we had considered the possibility of the formation of melt given these experimental conditions of pressure and

210 temperature, However, after extensive observation at the SEM, we did not find any microstructural evidence of melt pockets or pseudomorphs. We have added a sentence about that to the text (lines 242-243). In addition, according to thermodynamic modeling, the P-T conditions should be outside the melt-forming field.
* * *
215 ***Rev.2***: Line 276-282: please add sketches of the SC-SC' fabrics in Figs. 10a-c to better summarize these observations.

***Rev.2***: Line 303: whilst the SC' fabric is clear in Fig. 12c, Fig. 12a looks more an SC fabric. Please add sketches/annotations to highlight the fabric elements.

220 **Authors:** Here again, we decided to group the two previous comments together to avoid repetition. Following your advice, we have added annotations to several figures to highlight the fabric elements (Figs. 4a ; 6a-b ; 7a ; 12a-b-c). In the Mg-rich opx + plag samples, strain tends to produce mixed and connected fine-grained bands with C-type geometry in high-strain zones (e.g. Fig. 10a). In contrast, the Fe-rich opx + plag and amph + plag mixtures tend to form clusters and only locally connected amph-rich rich σ-tails at porphyroclasts, as S-C- (e.g. Fig 12a-b) or S-C'- (Fig. 12c) type geometries.

225
* * *
***Rev.2***: Line 312: I understand that the pole figures are plotted as one point per grain; please provide the total number of the plotted grains, the step size and the average grain size of amphibole, so that the reader can make their judgement on the data acquisition and processing routines. How many data points did you consider representative to define an individual "grain"?

230 Amphibole grain size in Fig. 13 looks < 1 micron, so I wonder whether many of you "grains" are actually individual data points that might encompass more than one single grain. Please clarify.

**Authors:** The reviewer has pointed out an important issue, we have added the data you mentioned (see caption of Figure 12 – line 1272). Then, we consider that 5 data points are required and are considered representative to define an individual grain. We have added a sentence about that to the methods section (line 151). With regard to the grain size in Figure 13, it is indeed extremely small. However, this figure is for the Mg-rich opx + plag assemblages. The pole figures in Figure 12 are associated only with the Fe-rich opx + plag assemblages. Unfortunately, none of our EBSD analyses performed on Mg-rich opx + plag assemblages were convincing, due in part to the extremely small grain sizes in the mixture zones. To clarify, we have indicated in the caption that the data presented in Figure 12 are only associated to the Fe-rich opx + plag assemblages (line 1269).
* * *
**Rev.2:** Line 360: is there any evidence of dislocation creep been potentially active in the strong phases? Do you have EBSD maps of porphyroclasts that could help understand this?

**Authors:** Unfortunately, none of our EBSD sessions on Mg-rich opx + plag assemblages were conclusive. This is most certainly due to the fact that we tried to study very fine grains within a narrow shear zone trapped between 2 very strong alumina pistons, thus making polishing very complex. The grain shapes of porphyropclasts do not suggest any deformation by crystal plasticity.
* * *
**Rev.2:** Line 494-497: this is a very interesting and plausible interpretation. But the follow up question is how did the $H_2O$ stored in the interior of strong porphyroclasts become available for the reactions? Did microfracturing play a role here? Any evidence?

**Authors:** The reviewer has made an impotant point here. Indeed, cracking in opx is very common : original Opx clasts are locally cut by brittle fractures which refines the grain size of strong porphyroclasts. The cracking will make the $H_2O$ available for reactions. We have added a sentence to the text to say this (lines 509-511).
* * *
**Rev.2:** Line 532: syn-kinematic mineral reactions are very important for the deformation of mafic systems also at higher metamorphic grades (see degli Alessandrini et al., 2017). Here you also document dehydration reactions and their role on deformation.

**Authors:** The reviewer is right. We have added this work to the reference list (line 550).

[revised manuscript text omitted]

1210

1215

[Figure]

a

Slip at sample/forcing
block interface

Goetze Criterion (GC)

Peak stress

Quasi steady-state

Weakening

Loading

differential stress (MPa)

shear strain (γ)

σ1

σ3

σ1

σ3

**Temperature:** 800-850-900°C
**Pressure:** 1 GPa -- **Strain rate:** $10^{-5}$ s$^{-1}$

Mixture **Mg-rich opx** + **plag** experiments

b

Short run-in

GC

OR47NM
$\tau_f$: 989

557NM
$\tau_f$: 577

559NM
$\tau_f$: 350

OR49NM
$\tau_f$: 800

OR41NM
$\tau_f$: 542

differential stress (MPa)

Mixture **Fe-rich opx** + **plag** experiments

d

GC

OR61NM
$\tau_f$: 897

533NM
$\tau_f$: 683

538NM
$\tau_f$: 563

532NM
$\tau_f$: 405

c

Long run-in

GC

OR51NM
$\tau_f$: 901

OR24NM
$\tau_f$: 866

OR38NM
$\tau_f$: 339

OR34NM
$\tau_f$: 126

differential stress (MPa)

shear strain (γ)

**amph** + **plag** and **pure amph** experiments

e

GC

OR23NM
$\tau_f$: 483

Amph+Plag

Pure Amph

OR18NM
$\tau_f$: 370

OR15NM
$\tau_f$: 147

Pure Amph

shear strain (γ)

[revised manuscript text omitted]